psychology

sexual selection, facial attractiveness, facial masculinity, facial hair, mate choice, human evolution

**Author for correspondence:**
Barnaby J. W. Dixson
e-mail: b.dixson@uq.edu.au

# A multivariate analysis of women's mating strategies and sexual selection on men's facial morphology

Tessa R. Clarkson[1], Morgan J. Sidari[1], Rosanna Sains[1], Meredith Alexander[1], Melissa Harrison[1], Valeriya Mefodeva[1], Samuel Pearson[1], Anthony J. Lee[2] and Barnaby J. W. Dixson[1]

[1]School of Psychology, University of Queensland, Brisbane, Queensland, Australia
[2]Division of Psychology, University of Stirling, Stirling, Scotland, UK

BJWD, 0000-0003-0911-1244

The strength and direction of sexual selection via female choice on masculine facial traits in men is a paradox in human mate choice research. While masculinity may communicate benefits to women and offspring directly (i.e. resources) or indirectly (i.e. health), masculine men may be costly as long-term partners owing to lower paternal investment. Mating strategy theory suggests women's preferences for masculine traits are strongest when the costs associated with masculinity are reduced. This study takes a multivariate approach to testing whether women's mate preferences are context-dependent. Women ($n = 919$) rated attractiveness when considering long-term and short-term relationships for male faces varying in beardedness (clean-shaven and full beards) and facial masculinity (30% and 60% feminized, unmanipulated, 30% and 60% masculinized). Participants then completed scales measuring pathogen, sexual and moral disgust, disgust towards ectoparasites, reproductive ambition, self-perceived mate value and the facial hair in partners and fathers. In contrast to past research, we found no associations between pathogen disgust, self-perceived mate value or reproductive ambition and facial masculinity preferences. However, we found a significant positive association between moral disgust and preferences for masculine faces and bearded faces. Preferences for beards were lower among women with higher ectoparasite disgust, providing evidence for ectoparasite avoidance hypothesis. However, women reporting higher pathogen disgust gave higher attractiveness ratings for bearded faces than women reporting lower pathogen disgust,

providing support for parasite-stress theories of sexual selection and mate choice. Preferences for beards were also highest among single and married women with the strongest reproductive ambition. Overall, our results reflect mixed associations between individual differences in mating strategies and women's mate preferences for masculine facial traits.

# 1. Introduction

Indirect and direct sexual selection has shaped the evolution of female preferences for male ornaments [1]. Indirect selection occurs when females select males displaying traits that reflect underlying genetic quality (e.g. health), while direct selection reflects preferences for traits that communicate tangible benefits like resources [1]. Sexual dimorphism in craniofacial morphology (i.e. facial masculinity) and men's facial hair are two possible targets of women's mate choices via indirect and direct sexual selection. Men's facial masculinity, which includes a protruding brow ridge, widened cheek bones, thick jawline and deeply set narrow eyes, is influenced by testosterone during fetal development [2], adolescence [3] and is fully developed in young adulthood [4]. As androgens may reduce immunity [5,6], investment in masculine traits may reflect a superior immune system [7]. Indeed, facial masculinity may reflect disease resistance [8,9] and immune response [10]. While some studies have not reported that facial masculinity is related to health or immunity [11,12], facial adiposity is also an index of male health [13] and an interaction between facial masculinity with facial adiposity predicts male immune response [14]. Alternatively, facial masculinity may communicate direct benefits, including resources and protection that enhance survival among mothers and their infants [15,16]. Facial masculinity is positively associated with physical strength [17,18], social assertiveness and fighting ability [19]. Increasing facial masculinity in stimuli experimentally enhances perceptions of male age, masculinity and dominance [20,21]. Thus, facial masculinity may provide information relating to men's underlying health and formidability that influences women's mating preferences.

Although both facial masculinity and facial hair require testosterone for their development and full expression in adulthood, total testosterone levels alone do not explain variation in the pattern, density and distribution of beardedness in men. Instead, beards develop as testosterone is synthesized into dihydrotestosterone via 5-alpha reductase activity within hair follicles [22,23]. Bearded faces are judged as looking older, more masculine, socially dominant and aggressive than clean-shaven faces [24–30]. Men with beards reported stronger feelings of masculinity [31], had higher serum testosterone that was, in turn, linked to higher social dominance [32], and held more stereotypical views of masculinity in heterosexual relationships than clean-shaven men [33,34]. However, unlike craniofacial masculinity, there is no evidence that beardedness reflects men's health or disease resistance [22,23]. Genome-wide association studies show that 74% of the variance in men's beard growth and density is owing to genetic factors [35]. While men with more physically masculine faces have greater upper body strength and fighting ability, there is no evidence that beardedness is associated with physical formidability or fighting performance [36]. Phylogenetic analyses among anthropoid primates suggest that beards function like visually conspicuous ornaments in male monkeys and apes in communicating age, social rank and social dominance within large social groups, multilevel social organizations and polygynous mating systems [37,38], potentially enhancing attractiveness to women by communicating direct benefits [39–41]. The causative effects of beards on judgments of masculinity, dominance and aggressiveness may explain why beards do not consistently enhance men's facial attractiveness [23]. However, beardedness is positively associated with male mating success within populations during periods when marriage markets are more male-biased and available partners were scarcer, potentially driving stronger intra-sexual competition [42]. Between-populations, facial hair is more common in larger cities where average incomes are low, health is high, sex ratios are more male biased and women's preferences for beardedness are higher [40,41].

In a similar vein, recent research also demonstrates that men with more masculine faces are more intra-sexually dominant and had higher mating success compared to their less masculine peers [43–46]. However, women's attractiveness judgments of facially masculine men vary considerably across studies, being higher for feminized faces in some [20,47], masculinized faces in others [23,48,49], while in several studies women's preferences were equivocal [50]. This variation may be explained by the social costs associated with masculinity in men as facially masculine men report higher preferences for short-term than long-term relationships [51,52], engage in more short-term than long-term relationships [46,53], state greater interest in extra-pair relationships [54], and engage in more extra-pair relationships than

less masculine men [55]. Furthermore, women accurately ranked photographs of male faces when judging their sexual infidelity using masculine facial cues [55–57]. Masculine men may be more aggressive than less masculine men [19], which has been linked to reduced preferences for facial masculinity [58]. As women invest significantly in pregnancy, lactation and child rearing [59], the higher emphasis on mating effort over paternal investment among masculine men may explain why less masculine men are preferred as more socially agreeable long-term partners.

According to mating strategies theory, women offset the costs of selecting less paternally investing men to secure indirect genetic benefits that enhance offspring fitness [60]. Thus, preferences for masculine traits are stronger among women of reproductive age when considering short-term than long-term relationships [61]. The ovulatory shift hypothesis states that women's preferences for masculine traits in men are strongest in and around the peri-ovulatory periods of the menstrual cycle, particularly when judging males for short-term sexual attractiveness [7]. Initial research reported stronger preferences for facial masculinity among women at the fertile phase than the luteal or menses phases of the menstrual cycle when judging general attractiveness and short-term attractiveness rather than long-term mate preferences [62]. However, these studies suffered from small sample sizes, between-subject designs [63] and employing questionnaires to determine women's current fertility rather than direct hormone measures [64]. Follow-up studies where women's fertility was measured via hormone assays together with correctly powered within-subject designs found no positive effects of high fertility on their attractiveness assessments of masculine facial features in men [65–67]. Likewise, attractiveness scores for beardedness are not stronger among women at the high than low fertile phases of the menstrual cycle in studies employing questionnaires [26,68–70] or when measuring women's reproductive hormones [65,71]. As women's preferences for men's facial hair are stronger when considering long-term relationships, it is possible that preferences for beardedness may not be expected to change with fertility. Nevertheless, taken together reproductive status over the menstrual cycle may not underpin differences in how women judge the mate value of men with well-developed masculine craniofacial traits or beards [72].

Plasticity in women's mate preferences for masculine traits in male faces may instead occur in concert with social and ecological factors. Natural selection in response to prevailing pathogenic conditions favoured an immune system that protects against parasites and pathogens [73] and a behavioural immune system, which encompasses abilities to detect and avoid pathogenic stimuli [74–76]. Individual differences in pathogen concern may underpin trade-offs in women's preferences for mates with higher health over reduced paternal investment [77]. Indeed, women who scored highly for disgust sensitivity rated masculine faces as more attractive than women with lower disgust sensitivity [78–80]. Furthermore, women living in cultures where life expectancy is compromised by disease and pathogens report the highest preferences for facial masculinity [81–84]. In the laboratory, exposure to cues of disease and infection causally increases the attractiveness of masculine faces to women ([85]; but see [86]). Parasite stress may also impact on how women perceive facial hair in potential mates. Reduced hirsutism in humans compared to other anthropoid primates may reflect natural selection for optimal body temperatures [87]. The ectoparasite avoidance hypothesis proposes that ancestral humans underwent additional loss of body hair as it lessened the potential for disease-carrying ectoparasites to proliferate [88–90]. Ectoparasites trigger forms of disgust that differ to disgust generated by pathogens [91] and are rated as disgusting, augment disgust responses and increase self-reported grooming behaviours [92–94]. The ectoparasite avoidance hypothesis suggests that sexual selection for reduced body hair may also have contributed to reduced hirsutism in humans [90]. While men's chest hair is judged to be sexually attractive among women from the UK and Cameroon, hairless chests are preferred among women from the USA, China, New Zealand, Finland, Brazil, Slovakia, Czechoslovakia and Turkey [95–104]. However, neither viewing photographic stimuli depicting diseases, illness and pathogens or responses to questionnaires measuring women's sensitivity to pathogens were linked to variation in women's attractiveness judgments of hair on the upper chest and abdomen in men [103,104]. Similarly, the attractiveness of facial hair was unchanged because of seeing images of pathogens or ectoparasites, although a positive relationship between disgust for pathogens and attractiveness ratings of beards was reported [86]. Whether or not this association between pathogen disgust and women's preferences for men's beardedness replicates remains to be determined.

Individual differences in self-reported attractiveness and mate value also impacts on whether women realize their stated mate preferences for masculine characteristics in men. Physically attractive women may be able to act upon preferences for masculine mates despite thier potentially lower paternal investment [105]. In support of this prediction, women with higher self-rated attractiveness and mate value judge masculine male faces as more attractive than women with lower self-reported mate value [106]. Drawing on mating strategies theory, Watkins [107] hypothesized that if women's preferences for

masculine partners reflect sexual selection for indirect genetic benefits (i.e. health), then women who desire to have children should have higher preferences for facial masculinity than women who do not desire offspring. Indeed, Watkins [107] reported that preferences for facial masculinity in men were highest among women currently in relationships who desired children. In contrast to facial masculinity, the positive effects of facial hair on ratings of social dominance and sexual maturity may explain why beards are judged as more attractive for long-term parentally investing relationships [25,26,39,49] rather than short-term relationships. Mothers with young infants and women with higher parity judged beards more favourably for parenting, masculinity and age, but lower for sexual attractiveness, than non-pregnant, nulliparous women [39]. Beardedness was also more common among fathers than non-fathers, suggesting that women's preferences for beards are higher for long-term partners with whom they have children than for short-term relationships [108]. Thus, women desiring children may judge beards to be more attractive for long-term relationships than women who do not desire to have children.

Sexual selection may operate on preferences for attractive traits through single preference functions for multiple sexual ornaments or multiple preferences for multiple ornaments [1]. As craniofacial shape and facial hair develop owing to different androgenic processes, they potentially communicate different aspects of male quality that are reflected in differing preference functions among women [23]. Thus, facial masculinity may provide salient information regarding men's health and physical strength that may augment male attractiveness as short-term partners among women selecting potential indirect genetic benefits when the costs of low paternal investment are mitigated [23]. By contrast, beardedness may not provide information regarding current male health and instead communicate long-term mate qualities such as maturity and social status [23]. Indeed, explicit ratings of age, masculinity and dominance rise linearly with facial hair density, so that facial hair reflecting 5–10 days of growth (i.e. stubble) receive intermediate ratings [25,26]. While women's preferences for clean-shaven and fully bearded faces differ over studies, studies using stimuli with a wider range of facial hair amounts reported that stubble is judged as most attractive [25,26,109,110]. In contrast to craniofacial masculinity, men can also groom, shape or remove their facial hair entirely, essentially increasing and decreasing their perceived masculinity at almost no cost biologically [40]. Beards amplify masculine facial features, particularly jaw size, such that less masculine faces are judged as more masculine and dominant when bearded than highly masculine clean-shaven male faces [21,28]. Variation in facial masculinity and beardedness also influence women's attractiveness ratings of male faces. Male faces manipulated to appear more feminine are rated as more attractive when lightly and heavily bearded, whereas highly masculine faces are judged as least attractive when full bearded [23]. These patterns in preferences have implications with regards how two sexually dimorphic androgen-dependent facial traits operate to enhance men's attractiveness. The divergent associations between indirect and direct quality components in facial masculinity and beardedness, respectively, suggest that sexual selection may have shaped different preference functions for each trait.

One way to expose any divergence in women's preference functions for men's facial hair and facial masculinity is through multivariate studies of individual differences in women's short-term and long-term mating orientation. To this end, the current study tests a series of hypotheses regarding how individual differences might explain variation in women's preferences for sexual dimorphism in male facial traits. With regards women's preferences for masculine facial shape, hypothesis 1 tested whether higher facial masculinity preferences occur among women with greater concerns of pathogenic infection, measured using the Three Domain Disgust Scale [111], which assesses pathogen, moral and sexual disgust. If preferences for facial masculinity reflect facultative trade-offs between paternal investment and genetic benefits, women who are high in pathogen disgust, but not moral or sexual disgust, should give higher ratings for facial masculinity [78–80]. Hypothesis 2 provided a further test regarding whether women's preferences for facial masculinity reflect selection for indirect genetic benefits, wherein women's reproductive ambition measured using the Desire for Pregnancy Subscale should be positively associated with their facial masculinity preferences [107]. We also attempted to directly replicate past results that reproductive ambition and preferences for facial masculinity occurs among women currently in relationships rather than single women [107]. Self-perceived mate value may moderate women's preferences for masculine characteristics in male partners, so that women of high self-perceived mate value state higher preferences for facial masculinity than women of lower mate value [105]. Hypothesis 3 tested this prediction using women's responses to the Self-Perceived Mate Value Scale [112]. As mating strategies theories propose that women's facial masculinity preferences should be highest when judging short-term than long-term relationships [61], we predicted that the positive associations between pathogen disgust and self-reported mate value outlined in hypotheses 1 and 3 would be stronger when judging stimuli for a short-term than a long-term relationship.

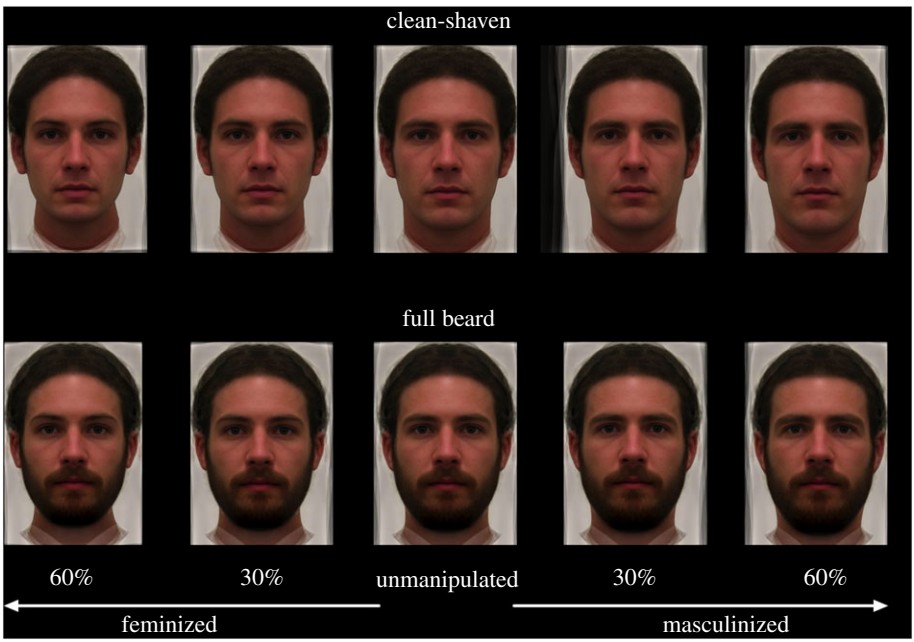

**Figure 1.** An example of the stimuli that were used in the current study. Stimuli depict composite faces comprised five males photographed with full beard (bottom row) and again when clean-shaven (top row). The composites were manipulated to appear 60% and 30% feminized, unmanipulated, and 30% and 60% masculinized.

With regards individual differences in women's attractiveness judgements of male facial hair, the ectoparasite avoidance hypothesis posits that reduced hirsutism evolved, in part, via mate choice as body hair may harbour disease-carrying ectoparasites. [86]. Hypothesis 4 tested whether women's preferences for beardedness were negatively associated with their disgust ratings of ectoparasites [86]. Alternatively, the parasite-stress handicap hypotheses suggest that sexually selected traits are costly to maintain and only higher-quality males may be able to withstand their detrimental effects on health [75]. Hypothesis 5 tested whether women reporting higher pathogen disgust stated higher preferences for facial hair than women reporting lower pathogen disgust [86]. In contrast to women's preferences for facial masculinity, women report stronger preferences for facial hair than clean-shaveness when judging long-term than short-term relationships and parenting skills [25,26,39,49]. Mothers reported stronger preferences for beardedness when judging parenting skills, but not attractiveness, compared to women without children [39] and women in long-term relationships with bearded men reported higher reproductive success than women in long-term relationships with non-bearded men [108]. Hypothesis 6 tested whether women with a greater desire for pregnancy preferred bearded stimuli when judging long-term rather than short-term relationships. Finally, social exposure during development and in daily life as an adult may influence women's judgements of male facial hair [40,109], whereby women in relationships with bearded men may prefer beards as a consequence of their visual diet [40]. Imprinting effects may also manifest among women who grew up with bearded fathers, resulting in strong preferences for beards in a potential partner [68,100]. Hypothesis 7 tested whether visual diet and imprinting effects were positively related to women's preferences for men's facial hair.

## 2. Methods

### 2.1. Facial hair stimuli

A total of 37 males (mean age = 27.86, s.d. 5.75 years), each of whom was ethnically Northern European were recruited when fully bearded (defined as four to eight weeks of untrimmed facial hair growth) and again when clean-shaven. Each participant posed for front posed photographs of their faces from the neck up with a neutral expression. All stimuli were collected in the same room under fluorescent lighting and at a standard 150 cm distance from the camera (figure 1).

## 2.2. Facial masculinity manipulation

We generated a set of controlled and standardized clean-shaven and bearded composite faces by placing 189 facial landmarks on all original unmanipulated faces. We then randomly selected sets of five males (clean-shaven and bearded versions) from the full sample of 37 males using EXCEL and combined their clean-shaven versions to make a clean-shaven facial composite and then combined their bearded versions to generate bearded facial composites. We repeated this process three times to generate three different composite bearded and clean-shaven faces. We then generated a composite male face and a composite female face using the same approach as above but employing a different collection of 40 female and 40 male European faces acquired from the website 3d.sk. Masculine facial shape was then adjusted using linear variation in masculinity and femininity from the male composite and the female composite which were adjusted in the bearded and clean-shaven facial composites at 30% and 60% feminized as well as 30% and 60% masculinized. Participants also rated the unmanipulated composite. This procedure is the most commonly used in facial attractiveness research [20,113] and alters composite facial dimorphism without changing texture or colour (figure 1).

## 2.3. Procedure

The study was developed on Qualtrics and administered on Amazon Mechanical Turk (MTurk). Participants began by providing informed consent to proceed and participate in the study. Participants were then shown three male composite faces that varied on five levels of masculinity, from very feminine (30%) to very masculine (60%) and were either bearded or clean-shaven. These faces were presented randomly to participants. In total, participants saw 30 (15 bearded and 15 clean-shaven) male faces. As faces appeared, participants were asked to rate how attractive they thought the males were for long-term and short-term relationship contexts. Their responses were collected using a 100-point Likert scale varying in 1 point increments where 0 = extremely unattractive and 100 = extremely attractive. Relationship contexts were defined using Little & Jones [114] as follows.

*Short-term relationship.* 'You are looking at the type of person who would be attractive in a short-term relationship. This implies that the relationship may not last a long time. Examples of this type of relationship include single date accepted on the spur of the moment, an affair within a long-term relationship, and possibly a one night stand' [114].

*Long-term relationship.* 'You are looking for the type of person who would be attractive in a long-term relationship. Examples of this type of relationship include someone you may want to move in with someone you may consider leaving a current partner to be with, and someone you may, at some point wish to marry (or enter a relationship on similar grounds as marriage)' [114].

## 2.4. Demographics

Each participant provided information pertaining to age (in years), whether they were male or female (with the option to select 'other'), whether they were in a relationship, their nationality, ethnic background and whether their father was present during their childhood, which were used in subsequent analyses of parental imprinting and social learning effects of preferences for men's beards. Sexual orientation was measured using the Kinsey scale.

## 2.5. Three domain disgust scale

The three domains of disgust scale were used to measure disgust sensitivity. The full survey comprises 21 questions that quantify disgust responses via Likert scales with seven points in which a score of one reflected 'not at all disgusting', and a score of seven represented 'extremely disgusting' [111]. The full set of 21 questions can be subdivided into three distinct areas of disgust: pathogen, moral and sexual disgust. Ratings showed strong internal consistency for moral disgust ($\alpha = 0.93$), sexual disgust ($\alpha = 0.82$) and pathogen disgust ($\alpha = 0.79$).

## 2.6. Ectoparasite avoidance scale

There is no known item scale that measures the emotional sensation felt by the thought or sight of skin-dwelling insects [91]. Thus, in the current study participants were presented with images of ectoparasites adapted from McIntosh *et al.* [86] and rated them on three sensations (disgust, afraid and creeped-out).

The stimuli taken from McIntosh *et al.* [86] to create this measure included images of burrowing ticks including: Australian paralysis tick (*Ixodes holocyclus*), sheep ticks (*Ixodes ricinus*), pubic louse (*Pthirus pubis*) and body louse (*Pediculus humanus humanus*). All photographs showed insects attached to hairs, burrowing into skin or on the skin surface and had been previously validated to elicit women's disgust responses over control images ($t_{98} = 26.11$, $p < 0.001$; $d = 3.31$) by McIntosh *et al.* [86]. Participants were asked to rate each 'To what extent does this make you feel: disgusted, creeped-out, or afraid'. Participants responded using Likert scales with seven points ranging from 1 (*not at all*) to 7 (*very much*). The internal consistency for the images was excellent for feeling disgusted ($\alpha = 0.94$), creeped-out ($\alpha = 0.97$) and afraid ($\alpha = 0.95$). These items were also highly correlated (all $rs \geq 0.62$, $p < 0.001$). Thus, we collapsed them into a singular variable hereafter referred to as ectoparasite avoidance ($\alpha = 0.96$).

## 2.7. Desire for pregnancy subscale

To measure pregnancy desire, participants completed the Pregnancy Desire Questionnaire, a revised six-item Likert scale from the larger eight-item Schaefer and Manheimer Desire for Pregnancy Subscale [107]. Evidence for the validity of this scale is reported in Watkins [107]. Participants responded using a 7-point Likert scale (with the anchors 1 = strongly disagree to 7 = strongly agree and 1 = extremely unhappy to 7 = extremely happy). This measure determines the participant's desire for a baby (e.g. I am looking forward to having a baby one day). One item of the six was reverse-scored (i.e. I do not want to have a baby at this time). Responses to item 22 'I try to keep from becoming pregnant' were removed for having poor internal consistency ($\alpha = 0.63$). After removal, the remaining items held good internal consistency ($\alpha = 0.82$).

## 2.8. Self-perceived mate value

Participants completed the four-item Mate Value Scale [112]. The four items in the scale were: (1) overall, how would you rate your level of desirability as a partner? (2) overall, how would members of the opposite sex rate your level of desirability as a partner? (3) overall, how do you believe you compare to other people in desirability as a partner? (4) Overall, how good of a catch are you? Each item was rated using a seven-point Likert scale from 1 = extremely low to 7 = extremely high. The four items had strong internal reliability ($\alpha = 0.90$).

## 2.9. Partner and father's level of facial hair

Participants stated the level of facial hair that was the most appropriate of 10 possible facial hair styles (0 = clean-shaven, 1 = stubble, 2 = moustache, 3 = goatee (without moustache), 4 = goatee (with moustache), 5 = sideburns, 6 = sideburns and moustache, 7 = moustache and soul patch, 8 = full beard (trimmed), 9 = full beard (bushy); electronic supplementary material, figure S1) when considering the typical facial hair for their partner or the level of facial hair their father had when recollecting their childhood. For our analyses, we created three categories: (i) the 'clean-shaven' category which included the percentage of men with no facial hair of any kind, (ii) the 'non-beard facial hair' category included the percentage of men in all classes of facial hair except clean-shaven and full beards (1–7), and (iii) the 'beard' category included the percentage of men with trimmed and bushy full beards (8 and 9).

## 2.10. Participants

Participants were recruited through the web-based marketplace research program Amazon Mechanical Turk (MTurk). By using MTurk, researchers can recruit large non-student samples [115]. To qualify for the current study, participants had to be female, sexually attracted to males and aged between 18 and 70. Participants were first screened for gender so that only females remained in the study. Participants were screened for sexual orientation. Sexual orientation can significantly attenuate female preferences for facial hair [100] and masculine facial characteristics [116–118]. Thus, we screened participants' sexual preferences using responses to the Kinsey Scale [119] and retained participants who reported being exclusively heterosexual to equally heterosexual and homosexual.

Of the total 1087 female participants that completed the survey, 919 participants aged 18–70 (mean = 37.47, s.d. = 12.09) remained for our final analyses after removing those who did not satisfy the selection criteria. The survey took approximately 20 min to complete and participants were compensated $1.00 USD for their time. Of the sample, 78% described themselves as White or

**Table 1.** Participant mean, standard deviations (s.d.) and ranges for the questionnaires.

| measure | mean | s.d. | range |
| --- | --- | --- | --- |
| pregnancy ambition | 3.02 | 1.41 | 0.83–5.83 |
| self-perceived mate value | 4.74 | 1.17 | 1.00–7.00 |
| ectoparasite avoidance | 5.22 | 1.48 | 0.00–6.00 |
| pathogen disgust | 4.13 | 1.06 | 0.140–6.00 |
| sexual disgust | 3.16 | 1.36 | 0.140–6.00 |
| moral disgust | 3.84 | 1.53 | 0.000–6.00 |

Caucasian, 9% were Black or African American, 6% were Asian, 5% were Hispanic and the remaining 2% were classified as Other. The majority of participants lived in the United States of America (98.1%). Participants were single (27.96%), dating (8.71%), married/committed relationship (61.92) or elected not to answer (1.41%). With regards sexual orientation, 78% were exclusively heterosexual, 12% were predominantly heterosexual and only incidentally homosexual, 4% were predominantly heterosexual but more than incidentally homosexual and 6% were equally heterosexual and homosexual. We provide the means, standard deviations and ranges for participant's responses to the questionnaires in table 1. Ethical clearance for this study was granted by the School of Psychology Ethics Review Panel at the University of Queensland (18-PSYCH-4G-12-JMC).

## 2.11. Statistical analyses

Analysis 1 used repeated measures ANOVA and Bayesian ANOVA where ratings for short-term and long-term attractiveness were the dependent variables. Facial masculinity (very low, low, neutral, high, very high) and facial hair (bearded, clean-shaven) were within-subject factors. Effect sizes in the models are eta square ($\eta^2$) and effect sizes for *post hoc* Bonferroni tests are Cohen's $d$. Bayesian analyses were undertaken to ascertain the presence or the absence of a hypothesized effect over the competing null effect. The Bayes factor ($BF_{10}$) provides an estimation of the strength of support a hypothesis receives relative to another competing hypothesis. A $BF_{10}$ of 1–3 is considered weak evidence, a $BF_{10}$ of 3–10 is considered moderate evidence and a $BF_{10}$ above 10 is considered strong evidence.

Analysis 2 was conducted using linear mixed effects modelling using the lme4 [120] and lmerTest [121] packages in R [122]. Ratings of attractiveness were the outcome variables in four separate models. All models included facial masculinity and facial hair as the stimulus-level predictors, as well as their interactions with participant-level variables. Participant-level predictors included in model 1 were pathogen, sexual and moral disgust, as well as ectoparasite avoidance, model 2 included mate value, model 3 included pregnancy ambition, while model 4 included father and partner beardedness. All continuous predictors were $z$-standardized at the appropriate group-level and dichotomous variables were given codes of −0.5 and 0.5. Random intercepts were specified for each participant, and each stimulus identity. Random slopes were specified maximally following recommendations in Barr *et al*. [123] and Barr [124]. Here, we report the fixed effects from each model; for full model specifications and results, including random effects, see the electronic supplementary material, S2.

# 3. Results

## 3.1. Analysis 1: the effect of facial hair and facial masculinity on women's short- and long-term attractiveness judgements

Facial hair had a significant main effect on women's short-term and long-term attractiveness that also received strong support in Bayesian analyses (table 2). Faces were more attractive when bearded than when clean-shaven for both short-term relationships, $t_{918} = 9.94$, $p < 0.001$, $d = 0.33$, and long-term relationships, $t_{918} = 10.22$, $p < 0.001$, $d = 0.34$. Facial masculinity also had a significant main effect on long- and short-term attractiveness ratings that received strong support in Bayesian analyses (table 2). Unmanipulated and highly masculinized faces were rated as more attractive than every other level of

**Table 2.** Repeated measures ANOVAs and Bayesian ANOVAS testing how masculinity (60% and 30% feminized, unmanipulated, 30% and 60% masculinized) and facial hair (clean-shaven and bearded) influence women's preferences for short- and long-term partners.

**repeated measures ANOVA**

| | short-term | | | | long-term | | | |
| --- | --- | --- | --- | --- | --- | --- | --- | --- |
| | $F$ | d.f. | $p$ | $\eta^2$ | $F$ | d.f. | $p$ | $\eta^2$ |
| facial hair | 98.78 | 1,918 | <0.001 | 0.012 | 104.44 | 1,918 | <0.001 | 0.015 |
| masculinity | 85.30 | 4,3672 | <0.001 | 0.004 | 140.29 | 4,3672 | <0.001 | 0.007 |
| facial hair × masculinity | 5.66 | 4,3672 | < 0.001 | 0.000 | 8.05 | 4,3672 | <0.001 | 0.000 |

**Bayesian repeated measures ANOVA**

| | short-term | | long-term | |
| --- | --- | --- | --- | --- |
| | $BF_{10}$ | error% | $BF_{10}$ | error% |
| facial hair | $6.737 \times 10^{106}$ | 1.386 | $6.132 \times 10^{115}$ | 0.966 |
| masculinity | $1.353 \times 10^{31}$ | 2.033 | $8.881 \times 10^{48}$ | 1.053 |
| facial hair × masculinity | $1.618 \times 10^{140}$ | 0.89 | $2.636 \times 10^{168}$ | 1.124 |

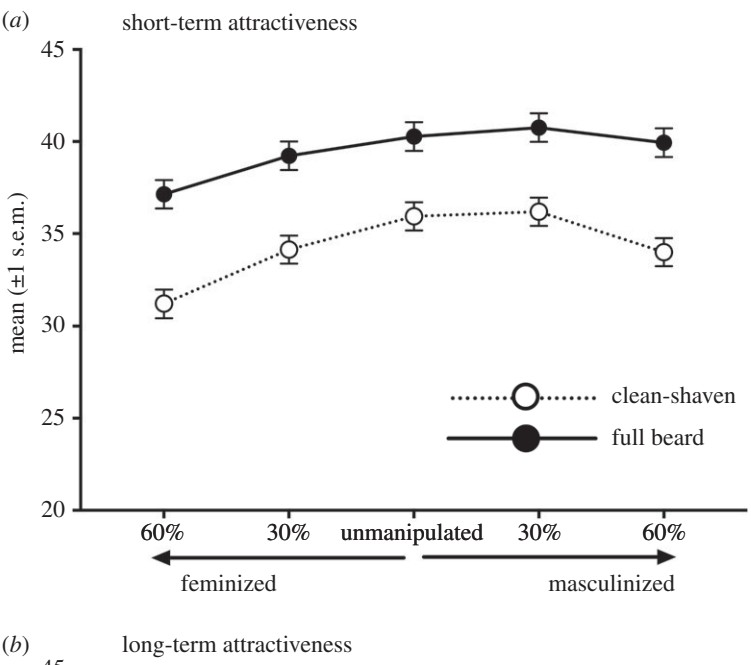

(a) short-term attractiveness

clean-shaven

full beard

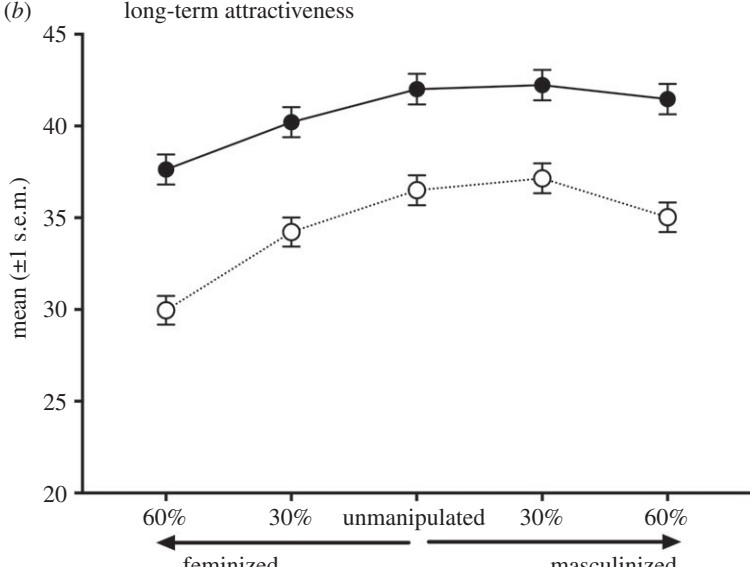

(b) long-term attractiveness

**Figure 2.** Mean ratings (±1 s.e.m.) for attractiveness when judging short-term (a) and long-term (b) relationships for bearded (black circles) and clean-shaven (white circles). The composites were manipulated to appear 60% and 30% feminized, unmanipulated, and 30% and 60% masculinized. Note that the full rating scale ranges from 0 to 100.

masculinity for both short-term, all $ts \leq 14.31$, all $ps < 0.001$, all $ds \leq 0.47$ and long-term conditions, all $ts \leq 18.51$, all $ps < 0.001$, all $ds \leq 0.61$. The most feminine faces were rated as least attractive for both short-term, all $ts \leq -8.63$, all $ps < 0.001$, all $ds \leq 0.29$, and long-term attractiveness, all $ts \leq -12.43$, all $ps < 0.001$, $ds \leq 0.41$. Ratings for neutral and high masculine faces did not differ significantly for ratings of short-term attractiveness, $t_{918} = 1.72$, $p = 0.860$ $d = 0.06$, and long-term attractiveness $t_{918} = 1.85$, $p = 0.642$, $d = 0.06$. Very masculine faces were rated significantly more attractive than the least masculine faces for short-term relationship context, $t_{918} = 8.63$, $p < 0.001$, $d = 0.29$ and long-term attractiveness, $t_{918} = 12.43$, $p < 0.001$, $d = 0.41$ (figure 2).

A significant facial masculinity × facial hair interaction was found for both long- and short-term relationship attractiveness that received strong support in Bayesian analyses (table 2). Ratings were significantly higher for bearded faces compared to clean-shaven faces within every level of masculinity for short-term, all $ts \geq 7.32$, all $ps < 0.001$, all $ds \leq 0.24$, and long-term all $ts \geq 7.49$, all $ps < 0.001$, all $ds \leq 0.25$, relationship contexts. However, the strength of facial hair on attractiveness ratings differed owing to the degree of facial masculinity. Within bearded faces, attractiveness ratings for masculine and very masculine faces were significantly greater than feminine and very feminine for

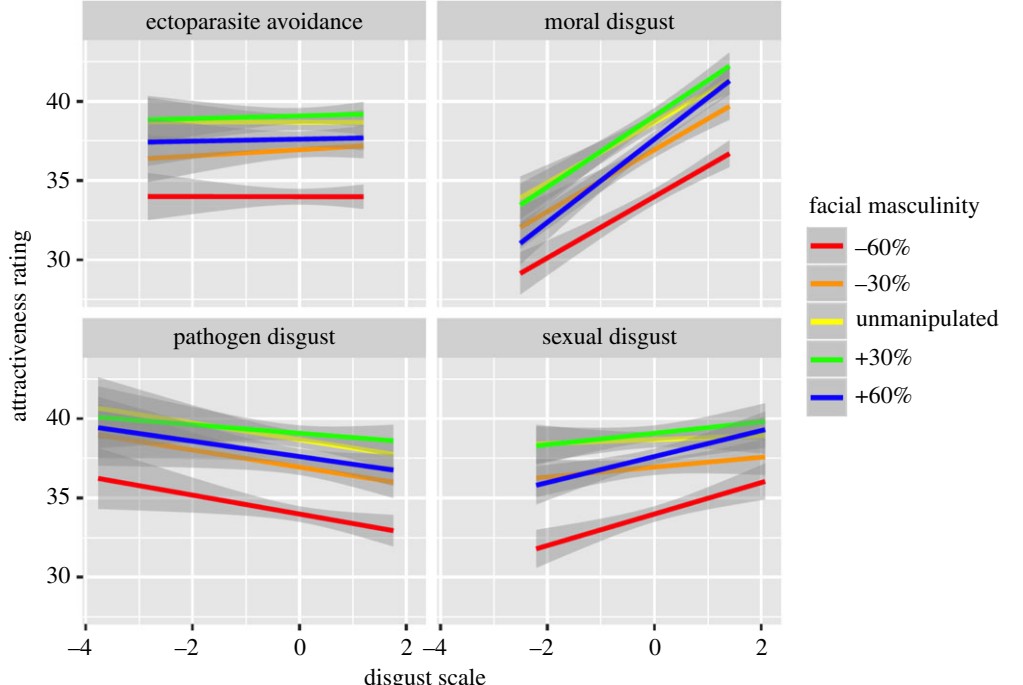

**Figure 3.** The associations between women's ectoparasite, moral, pathogen and sexual disgust and their attractiveness ratings for male facial masculinity. The lines represent the different levels of facial masculinity. The data represent regression lines (±95% confidence interval). The full rating scale ranges from 0 to 100.

short-term, all $ts \geq 4.50$, all $ps < 0.001$, all $ds \leq 0.15$, and long-term, $ts \geq 3.37$, $ps \geq 0.034$, all $ds \leq 0.11$, relationships. Unmanipulated faces were more attractive than very feminine faces for short-term, $t_{918} = 9.18$, $p < 0.001$, $d = 0.30$, and long-term, $t_{918} = 11.64$, $p < 0.001$, $d = 0.38$, and for feminine for long-term, $t_{918} = 4.80$, $p < 0.001$, $d = 0.16$, but not for short-term, $t_{918} = 3.06$, $p = 0.101$, $d = 0.10$, relationships. Preferences for bearded faces with unmanipulated, masculine and very masculine faces did not differ significantly for short-term, all $ts \leq 2.40$, $ps \geq 0.748$, all $ds \leq 0.08$, and long-term relationships $ts \leq 1.43$, $ps = 1.000$, all $ds \leq 0.05$. Clean-shaven unmanipulated and masculine faces were rated as more attractive than very feminine, slightly feminine and very masculinity, short-term, all $ts \geq 5.30$, all $ps < 0.001$, all $ds \leq 0.18$, and long-term, $ts \geq 3.92$, all $ps < 0.01$, all $ds \leq 0.13$, relationships. Preferences for unmanipulated faces were significantly higher than very masculine faces for short-term, $t = 5.69$ $p < 0.001$, $d = 0.19$, and long-term, $t = 3.92$, $p < 0.01$, $d = 0.13$, relationships. Very masculine faces were rated more attractive than very low masculinity for short-term, $t_{918} = 8.17$, $p < 0.001$, $d = 0.27$, and long-term, $t_{918} = 13.50$, $p < 0.001$, $d = 0.45$, relationships (figure 2).

## 3.2. Analysis 2: individual differences in facial masculinity and beardedness preferences

Across all of the following models, we found significant main effects of beardedness and facial masculinity on attractiveness ratings in line with the ANOVA analyses above. We also found a significant main effect of short-term versus long-term attractiveness in three of the four models, which suggests that participants gave higher ratings when considering long-term attractiveness compared with short-term attractiveness.

Women's preferences for facial masculinity were unrelated to their self-reported pathogen disgust, providing no support for hypothesis 1 that higher facial masculinity preferences occur among women with higher concerns of pathogenic infection (figure 3). Indeed, only moral disgust was significantly associated with preference for facial masculinity, such that participants higher in moral disgust rated facial masculinity as more attractive than participants who reported lower moral disgust (table 3).

There was also no significant association between women's preferences for facial masculinity and their self-reported desire to become pregnant (table 4). This provides no support for hypothesis 2, which proposed women's preferences for facial masculinity reflect selection for indirect genetic benefits. Women reporting high self-perceived mate value have been previously reported to state

**Table 3.** The fixed effect estimates for the model including pathogen, sexual, and moral disgust, and ectoparasite avoidance. (*$p < 0.05$; **$p < 0.01$; ***$p < 0.001$.)

| | estimate (s.e.) | t-value (approx. d.f.) | p-value |
|---|---|---|---|
| intercept | 37.26 (0.90) | 41.29 (10.29) | <0.001*** |
| short/long-term | 0.76 (0.34) | 2.22 (914.04) | 0.027* |
| pathogen disgust | −1.55 (0.93) | −1.66 (60.19) | 0.102 |
| short/long-term | −0.54 (0.43) | −1.26 (914.04) | 0.207 |
| sexual disgust | 0.25 (0.82) | 0.31 (318.69) | 0.756 |
| short/long-term | 1.34 (0.40) | 3.36 (914.04) | 0.001** |
| moral disgust | 2.49 (0.77) | 3.25 (334.26) | 0.001** |
| short/long-term | −0.05 (0.38) | −0.12 (914.04) | 0.902 |
| ectoparasite avoidance | 0.31 (0.80) | 0.39 (263.34) | 0.695 |
| short/long-term | −0.18 (0.39) | −0.45 (914.04) | 0.652 |
| facial masculinity | 1.48 (0.12) | 12.41 (914.29) | <0.001*** |
| short/long-term | 0.63 (0.13) | 4.86 (12035.80) | <0.001*** |
| pathogen disgust | 0.08 (0.15) | 0.55 (914.29) | 0.585 |
| short/long-term | −0.17 (0.16) | −1.02 (12035.80) | 0.309 |
| sexual disgust | −0.19 (0.14) | −1.35 (914.29) | 0.178 |
| short/long-term | −0.02 (0.15) | −0.16 (12035.80) | 0.871 |
| moral disgust | 0.31 (0.13) | 2.38 (914.29) | 0.017* |
| short/long-term | 0.06 (0.14) | 0.42 (12035.80) | 0.674 |
| ectoparasite avoidance | −0.04 (0.14) | −0.30 (914.29) | 0.767 |
| short/long-term | 0.02 (0.15) | 0.13 (12035.80) | 0.894 |
| beardedness | 5.66 (0.50) | 11.33 (914.30) | <0.001*** |
| short/long-term | 0.96 (0.45) | 2.15 (914.76) | 0.032* |
| pathogen disgust | 1.28 (0.63) | 2.04 (914.30) | 0.042* |
| short/long-term | 0.49 (0.56) | 0.88 (914.76) | 0.381 |
| sexual disgust | −4.31 (0.58) | −7.41 (914.30) | <0.001*** |
| short/long-term | −0.32 (0.52) | −0.62 (914.76) | 0.533 |
| moral disgust | 2.05 (0.55) | 3.75 (914.30) | <0.001*** |
| short/long-term | 0.00 (0.49) | −0.01 (914.76) | 0.992 |
| ectoparasite avoidance | −1.31 (0.56) | −2.32 (914.30) | 0.021* |
| short/long-term | −0.60 (0.50) | −1.19 (914.76) | 0.235 |

higher preferences for facial masculinity than women of lower mate value. However, we found no support for hypothesis 3, as there were no significant associations between women's preferences for male facial masculinity and their self-perceived mate value (table 5). To test whether reproductive ambition and facial masculinity preferences were significantly higher among women currently in relationships than women not in relationships, we re-ran these analyses to include women's current relationship status (married/committed relationship versus single/dating) as an additional factor. This analysis revealed no significant associations among reproductive ambition, relationship status and facial masculinity preferences and is reported in the electronic supplementary material, S1 and table S1.

All four disgust measures were significantly associated with preference for beardedness (table 3). In support of hypothesis 4, women's preferences were negatively associated with their disgust towards ectoparasites and in support of hypothesis 5 women's preferences for beards were positively associated with their self-reported pathogen disgust (figure 4). Participants high in sexual disgust showed a decreased preference for beardedness, while participants high in moral disgust showed stronger preferences for beards (table 3). There was a significant interaction between relationship type

**Table 4.** The fixed effect estimates for the model including pregnancy ambition. (* $p < 0.05$; *** $p < 0.001$.)

|  | estimate (s.e.) | t-value (approx. d.f.) | p-value |
|---|---|---|---|
| intercept | 37.26 (0.90) | 41.28 (10.74) | <0.001*** |
| short/long-term | 0.76 (0.34) | 2.21 (917.00) | 0.028* |
| pregnancy ambition | 0.39 (0.70) | 0.56 (479.35) | 0.577 |
| short/long-term | 0.33 (0.34) | 0.97 (917.00) | 0.332 |
| facial masculinity | 1.48 (0.12) | 12.39 (917.37) | <0.001*** |
| short/long-term | 0.63 (0.13) | 4.86 (12213.97) | <0.001*** |
| pregnancy ambition | −0.06 (0.12) | −0.51 (917.37) | 0.613 |
| short/long-term | 0.04 (0.13) | 0.35 (12213.97) | 0.73 |
| beardedness | 5.66 (0.52) | 10.96 (917.00) | <0.001*** |
| short/long-term | 0.96 (0.45) | 2.16 (917.13) | 0.031* |
| pregnancy ambition | −0.37 (0.52) | −0.72 (917.00) | 0.471 |
| short/long-term | 0.55 (0.45) | 1.23 (917.13) | 0.219 |

**Table 5.** The fixed effect estimates for the model including mate value. (* $p < 0.05$; *** $p < 0.001$.)

|  | estimate (s.e.) | t-value (approx. d.f.) | p-value |
|---|---|---|---|
| intercept | 37.26 (0.90) | 41.28 (10.75) | <0.001*** |
| short/long-term | 0.76 (0.34) | 2.21 (917.00) | 0.027* |
| mate value | −0.34 (0.71) | −0.49 (299.93) | 0.626 |
| short/long-term | 0.63 (0.34) | 1.84 (917.00) | 0.066 |
| facial masculinity | 1.48 (0.12) | 12.39 (917.00) | <0.001*** |
| short/long-term | 0.63 (0.13) | 4.92 (50539.01) | <0.001*** |
| mate value | −0.07 (0.12) | −0.57 (917.00) | 0.569 |
| short/long-term | −0.06 (0.13) | −0.44 (50539.01) | 0.663 |
| beardedness | 5.66 (0.52) | 10.97 (916.99) | <0.001*** |
| short/long-term | 0.96 (0.45) | 2.16 (917.00) | 0.031* |
| mate value | −0.77 (0.52) | −1.49 (916.99) | 0.137 |
| short/long-term | −0.83 (0.45) | −1.86 (917.00) | 0.063 |

and sexual disgust, such that there was a positive relationship between sexual disgust and attractiveness ratings only when participants were considering long-term attractiveness (figure 5).

We did not find any association between pregnancy ambition and preference for beardedness when judging either long-term relationships, providing no support for hypothesis 6 that preferences for beardedness should be higher among women with greater pregnancy ambition when judging long-term rather than short-term relationships. However, when we re-ran our analyses to include women' current relationship status (married/committed relationship versus single/dating) as an additional factor, there was a significant three-way interaction between facial hair, relationship status and pregnancy ambition (electronic supplementary material, S1 and table S1). This reflects that preferences for clean-shaven faces were positively associated with reproductive ambition among single women, while preferences for clean-shaven faces were negatively associated with reproductive ambition among partnered women (figure 6). Preferences for beardedness were positively associated with reproductive ambition among partnered and single women, with attractiveness ratings being higher overall for single than partnered women (figure 6).

From our total sample of 919, 662 women reported being in a relationship and reported the degree of beardedness in their partners. Thus, 194 (29%) of male partners were clean-shaven, 320 had non-bearded facial hair (48%) and 148 (22%) had full beards. In support of hypothesis 7, the degree of facial hair

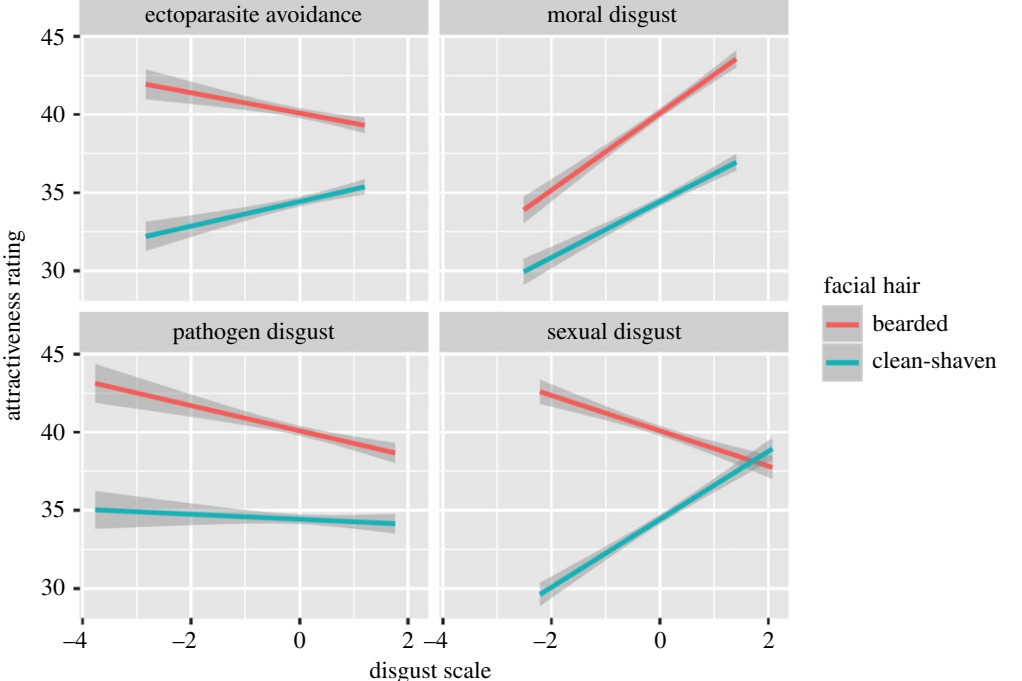

**Figure 4.** The associations between women's ectoparasite, moral, pathogen and sexual disgust and their attractiveness ratings for male beardedness when judging bearded faces (red line) and clean-shaven faces (green line). Data show regression lines (±95% confidence interval). Note that the full rating scale ranges from 0 to 100.

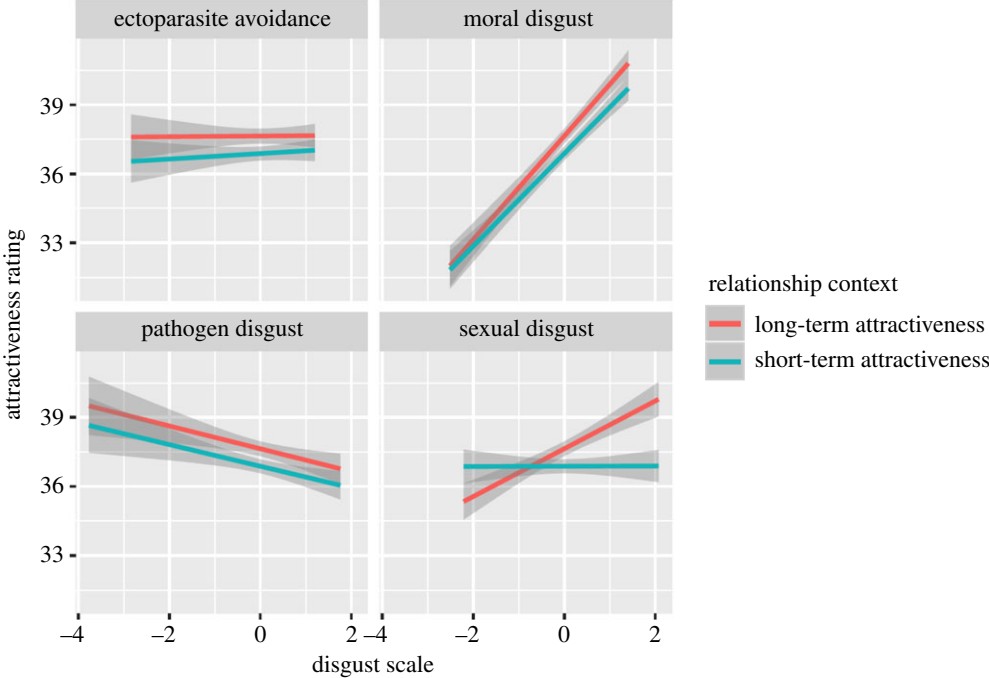

**Figure 5.** The associations between ectoparasite, moral, pathogen and sexual disgust and women's attractiveness ratings when judging a short-term relationship (green line) and a long-term relationship (red line). Data show regression lines (±95% confidence interval). Note that the full rating scale ranges from 0 to 100.

women reported in their partners was associated with their preferences for beardedness (table 6), such that women with partners with beards rated bearded faces as more attractive for both short-term or long-term attractiveness (table 6). In total, 754 women reported their father's beardedness in childhood, of which 362 (48%) were clean-shaven, 311 had non-bearded facial hair (41%) and 81 (11%) had full beards. However, there was no association between father's beardedness and women's

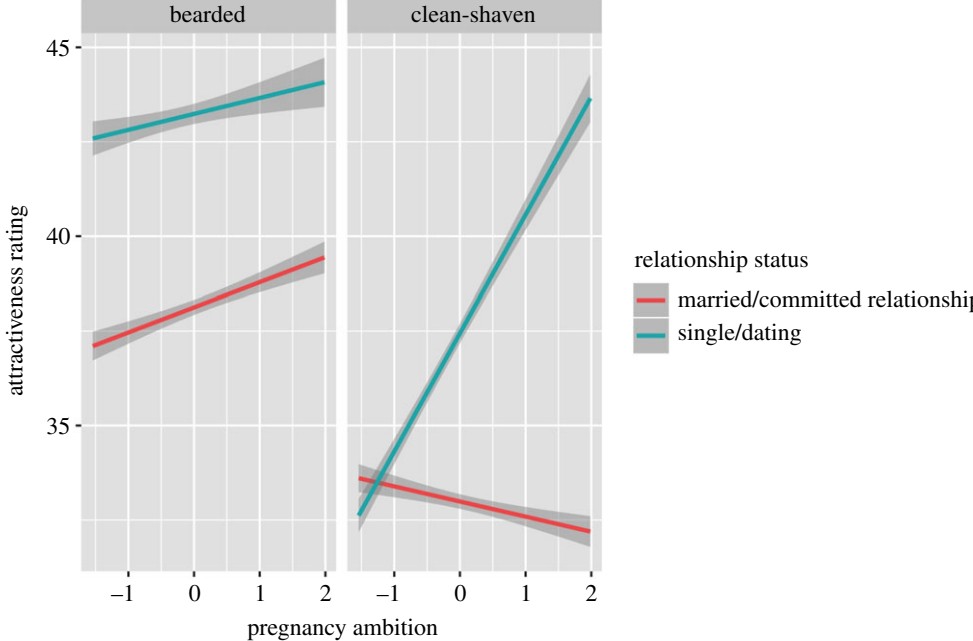

**Figure 6.** The associations between reproductive ambition and attractiveness ratings for clean-shaven and bearded faces among women in long-term/married relationships (red line) and women who were single/dating (green line). Data show regression lines (±95% confidence interval). Note that the full rating scale ranges from 0 to 100.

**Table 6.** The fixed effect estimates for the model including father and partner beardedness. (**$p < 0.01$; ***$p < 0.001$.)

| | estimate (s.e.) | *t*-value (approx. d.f.) | *p*-value |
|---|---|---|---|
| intercept | 36.28 (1.18) | 30.68 (9.06) | <0.001*** |
| short/long-term | 0.60 (0.35) | 1.72 (546.99) | 0.087 |
| father beardedness | −0.22 (0.89) | −0.24 (252.30) | 0.809 |
| short/long-term | 0.64 (0.35) | 1.83 (546.99) | 0.067 |
| partner beardedness | −0.12 (0.86) | −0.14 (513.32) | 0.893 |
| short/long-term | 0.63 (0.35) | 1.80 (546.99) | 0.073 |
| facial masculinity | 1.56 (0.15) | 10.45 (547.35) | <0.001*** |
| short/long-term | 0.63 (0.17) | 3.77 (7715.90) | <0.001*** |
| father beardedness | −0.28 (0.15) | −1.91 (547.35) | 0.056 |
| short/long-term | −0.11 (0.17) | −0.67 (7715.90) | 0.505 |
| partner beardedness | 0.25 (0.15) | 1.66 (547.35) | 0.097 |
| short/long-term | −0.06 (0.17) | −0.39 (7715.90) | 0.696 |
| beardedness | 5.15 (0.64) | 8.04 (546.99) | <0.001*** |
| short/long-term | 0.88 (0.51) | 1.71 (547.01) | 0.088 |
| father beardedness | 0.21 (0.64) | 0.34 (546.99) | 0.737 |
| short/long-term | 0.82 (0.51) | 1.62 (547.01) | 0.107 |
| partner beardedness | 5.67 (0.64) | 8.93 (546.99) | <0.001*** |
| short/long-term | 1.35 (0.51) | 2.66 (547.01) | 0.008** |

preferences for beards, providing no support for imprinting effects on women's facial hair preferences (table 6).

When we included age in our models, facial masculinity preferences were lowest among younger women and increased with age ($t = 3.483$, $p < 0.001$), but were not significant for preferences for

beardedness ($t = -1.841$, $p = 0.065$). However, there were no significant associations between age, preferences for facial masculinity or beardedness and variation in responses to any of the scales (see the electronic supplementary material, S2).

## 4. Discussion

Craniofacial masculinity and beardedness rely on different androgenic processes during development for their expression in adulthood, so that beards and facial shape can vary independently between individuals [23,28], potentially reflecting different mechanisms of sexual selection via female mate choice influencing their expression [23]. To test how variation in facial masculinity and beardedness determine judgements of male attractiveness, we measured women's preferences for male faces manipulated to produce five levels of facial masculinity (30% and 60% feminized, unaltered, 30% and 60% masculinized) in composite faces representing the same men when fully bearded and clean-shaven. Women rated beards higher for attractiveness compared with clean-shaven faces, particularly when judging long-term than short-term relationships. Past research has reported largely equivocal preferences for beardedness among women [39], partly owing to the degree of facial hair presented in the stimuli whereby light facial hair or 'stubble' is more attractive than full beards and clean-shaven faces in some studies [23–26,109]. However, our finding that women judged beards as more attractive for long-term than short-term relationships supports several previous studies [23,25,26,39,49]. Women's preferences were higher for slightly masculine and highly masculine faces than faces with lower masculinity, especially when judging long-term rather than short-term sexual attractiveness. While this pattern is the opposite to that reported in some of the past literature on women's facial masculinity preferences [61], preferences peaked at intermediate levels of masculinity (unmanipulated and +30%) rather than the most masculine faces, supporting previous research that also used stimuli in which masculinity varied incrementally [125]. This level of masculinity may reflect an optimal combination of masculine and feminine features that enhance aesthetic facial attractiveness in men [25,26]. In previous research, experimentally manipulating masculine facial cues impacts on the attractiveness of facial hair, so that the lower attractiveness judgements in the extremes of facial femininity and masculinity were attenuated when faces were bearded than when clean-shaven [23,28]. In the current study, the lower attractiveness judgements women ascribed to highly feminized and highly masculinized compared to intermediate levels of facial masculinity were less pronounced in bearded than clean-shaven faces, possibly because beards mask the unattractive aspects of facial morphology that are overemphasized in the most pronounced manipulations [25,26].

The potential for men's facial and body hair to allow disease-carrying ectoparasites to proliferate has led to two competing hypotheses regarding how sexual selection has shaped hirsutism [86]. Handicap theories of sexual selection suggest beards may augment male attractiveness via signalling an individual's ability to withstand the costs of ectoparasites [86]. Alternatively, the ectoparasite avoidance hypothesis proposes that reduced body hair in humans was elaborated upon by sexual selection as mating with less hirsute individuals would have lessened the chance of intra-individual transmission of diseases carried by ectoparasites [87–89]. Past research did not report women's preferences for clean-shaven faces or hairless chests were higher in countries with higher pathogen levels [40,41,103] or following exposure to cues of pathogens or ectoparasites [86,104]. However, in support for hypothesis 4 in the current study, women's disgust ratings of ectoparasites were negatively associated with preferences for men's beards. We also found a significant negative association between women's sexual disgust and their attractiveness ratings for facial hair, which is consistent with past research [86]. Associations between low sexual disgust and higher attractiveness ratings for men's masculine facial traits, potentially including beards [49], might reflect aspects of female sex drive underpinning preferences [126,127]. In previous research, attractiveness ratings of male beards were highest among women with less restricted global sociosexualities [49], and higher pathogen disgust [86]. In the current study, we also found a positive, albeit weak, association between women's attractiveness ratings for male beards and their self-reported pathogen disgust, providing some support for hypothesis 5. This could be interpreted as evidence that facial hair is preferred as a marker of health among women with high pathogen concerns, or that facial hair masks areas of the face that would communicate ill health. However, the positive interaction between preferences for facial hair and pathogen disgust was only significant in models that included the three other disgust measures. Thus, we interpret this finding with caution until further replications are published. Importantly, women's pathogen disgust and ectoparasite disgust follow opposite directions

concerning attractiveness ratings of male facial hair, suggesting selection for different preference functions. To our knowledge, this provides the first supporting evidence for the ectoparasite avoidance hypothesis for women's preferences when judging male hirsutism. Given that several studies did not support the ectoparasite avoidance hypothesis [86,103,104], we again urge caution in interpreting our findings until further replications have been undertaken.

In contrast to past research, we found no support for our hypotheses 1, 2 and 3 that facial masculinity is more attractive among women who report high pathogen disgust [78,79], high reproductive ambition among both partnered and single women [107] or high mate value [105]. Our null findings for context-specific associations and mate preferences in women for male facial masculinity complement recent studies. Thus, studies among female identical and non-identical twins reported that genetic variation explained 38% of the variation in attractiveness judgements of male facial masculinity, while pathogen disgust, sociosexuality and fertility explained less than 1% of the variance [128]. Recent experimental research has also called into question whether women's mate choices for males with more masculine faces differ under conditions wherein short-term mating strategies and possible indirect genetic benefits to offspring fitness may be gained [16,72]. For example, initial research reported that facial masculinity was preferred among women at the fertile phase of the menstrual cycle compared to other points of the cycle, particularly when making judgements of sexual attractiveness and short-term attractiveness [62]. However, recent research employing endocrine measures to characterize fertility have not reported stronger preferences for facial masculinity when fertility is highest [65–67]. Instead, preferences may shift owing to changes in endocrine status as women transition to motherhood or as a function of social expectations in partners changing over the life course [39]. Recent cross-cultural research reported that facial masculinity is most attractive among women from societies with higher urban development and lower pathogens [129,130], and one study reported no changes in facial masculinity following exposure to stimuli depicting pathogens [86]. Nevertheless, future replication is required to determine how robust pathogen disgust is in maintaining variation in mate preferences among women for facial masculinity.

In the current study, participants who scored high on moral disgust reported the highest overall ratings of attractiveness. Facial masculinity also interacted with self-reported moral disgust for attractiveness ratings, such that women's attractiveness ratings of masculine faces increased with increasing moral disgust. This replicated a past study that also reported a positive association between moral disgust and women's attractiveness ratings of male masculine facial features [86] and potentially reflects preferences for males with greater political conservatism [131,132]. However, we note that our study was very highly powered and the size of this effect was small, therefore despite statistical significance, our finding may have limited biological significance. Women's preferences for beardedness also increased with women's self-reported moral disgust, which may also reflect associations between beardedness and political conservatism [133] and traditional views regarding masculinity in heterosexual relationships ([33,34]; but see [134]). Further research on hegemonic masculinity and men's decisions to wear full beards and which social and political factors contribute to variation in women's mate preferences for facial hair in men would be an important addition to this literature.

Unlike facial masculinity, beardedness can easily be altered or removed entirely through grooming practices. Men's grooming decisions may not simply reflect variation in trends in fashion, but may be influenced by social, economic and ecological factors. Analyses of facial hair fashions across populations reveals that the frequency of facial hair is higher in bigger cities from countries with high health, low average incomes and where women's preferences for facial hair are highest [40]. Further, analyses of facial hair fashions in London from 1842 to 1972 reported that facial hair was more common among men during the years when the sex ratio in the potential marriage market was more male biased [42]. In another cross-cultural analysis, it was revealed that women's attractiveness judgements of male beards and body hair are higher in countries with more male-biased sex ratios [41]. Thus, social exposure to facial hair might influence mate choices for beards in partners, such that preferences for men's facial hair may be positively associated with the degree of beardedness in women's current partners [68,100,109] and their father's facial hair during childhood [68,100]. Hypothesis 7 tested these predictions, and while we found no associations between women's attractiveness ratings for beards and beardedness in their fathers', women in relationships with bearded men had stronger preferences for bearded faces. This may reflect that social exposure to beardedness influences the strength of women's preferences for facial hair [135] or that preferences for beardedness has an influence on women's choice of beardedness in their actual partners [26]. This pattern in preferences was also stronger when women made long-term rather than short-term

judgements of attractiveness. Past research has shown that ratings of long-term attractiveness and fatherhood were stronger for a bearded than clean-shaven appearance and these judgements were associated with women's reproductive success [39,108]. Interestingly, we found no support for hypothesis 6 that women with greater reproductive ambition would have stronger preferences for beardedness when judging long-term rather than short-term relationships. However, when we included women's current relationship status in the analyses, single and married women who desired to have a child rated beards as more attractive than women who did not desire to have child, although single women's ratings were higher overall than married women. Preferences for clean-shaven faces were positively associated with reproductive ambition among single women, while preferences among married women were negatively associated with reproductive ambition. Future research assessing whether beardedness is positively associated with male paternal investment and consequently offspring fitness would be valuable.

Although the current study included a large sample of female raters and composite male stimuli that manipulated facial masculinity and beardedness while controlling for various confounds that occur in natural stimuli (e.g. symmetry and blemishes), there are some important shortcomings to our study that should be noted. For instance, our sample included some women outside of the reproductive age-range typically used for quantifying mate preferences for masculine characteristics, so that reproductive status may have impacted on women's stated mate preferences. Preferences for facial hair are stronger among post-menopausal than pre-menopausal women [68] and among women with children than women without children [39]. Unfortunately, we did not collect data regarding our participants' reproductive status or parity, and we acknowledge that these differences between our participants may have influenced our results. However, we also note that the inclusion of participant's age in our models did not alter our results. Our study employed a multivariate approach to assessing human mate preferences, which has been successfully undertaken in previous research on men's preferences for female morphology [136,137] and women's preferences for men's morphology [138,139]. Our sample of participants is restricted to people from the USA and we hope this study inspires further replication across other populations, a possibility that can be realized via collaboration through established research networks [140]. For now, our findings provide mixed evidence that individual differences maintain variation in women's mate preferences for masculine secondary sexual facial traits in men.

Ethics. All participants gave informed consent to participate in this research and the study received ethical clearance from the University of Queensland's Behavioural and Social Sciences Ethical Review Committee and the School of Psychology's Ethics Review Panel (18-PSYCH-4G-12-JMC).

Data accessibility. This article does not contain any additional data.

Authors' contributions. B.J.W.D., M.J.S. and S.P. coordinated the study; T.R.C., B.J.W., R.S., M.A., M.H. participated in the design of the study; T.R.C., M.J.S., V.M., A.J.L. and B.J.W.D. carried out the statistical analyses; T.R.C., M.J.S., V.M., A.J.L. and B.J.W.D. drafted the manuscript; T.R.C., R.S., M.A. and M.H. collected data. All authors gave final approval for publication.

Competing interests. We declare we have no competing interests.

Funding. This study was supported by a University of Queensland Postdoctoral Fellowship to B.J.W.D.

Acknowledgements. We thank all the participants who completed our study. We are also grateful for the comments from the handling editor and two anonymous reviewers.

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
