## [Reviewer comments · Royal Society Open Science]

Review History

RSOS-191209.R0 (Original submission)

Review form: Reviewer 1

Is the manuscript scientifically sound in its present form?

Yes

Are the interpretations and conclusions justified by the results?

No

Is the language acceptable?

Yes

Do you have any ethical concerns with this paper?

No

Have you any concerns about statistical analyses in this paper?

Yes

Recommendation?

Accept with minor revision (please list in comments)

Comments to the Author(s)

The manuscript reports the results of a study investigating the associations between women's preferences for facial hair and for masculinised / feminised faces, along with various measures such as the women's self-reported disgust and desire for pregnancy. The manuscript contains a lot of interesting data and a helpfully large sample size, although I do have a number of questions, below.

1. The complete summary of the results, as provided in the Abstract, is as follows: "We found no associations between pathogen disgust, self-perceived mate value or reproductive ambition and facial masculinity preferences. However, the association between women's attractiveness ratings and their feelings of disgust for ectoparasites was negative, providing evidence for ectoparasite avoidance hypothesis". There is no mention of any of the other results. This creates the misleading impression that against a background of null findings, the one stand-out result was the association between attractiveness ratings and ectoparasite disgust. This is not true: for example, higher moral disgust was also associated with increased preference for bearded faces, which does not help the ectoparasite avoidance hypothesis. The Abstract should be adjusted to point out that the picture is muddier than that currently portrayed. (The authors might also choose to consider whether there are also other interesting results that could be incorporated in the Abstract, space permitting.)
2. As far as I am aware, there is no evidence that men's facial hair patterns (density, growth) are associated with their testosterone levels. This is probably worth stating explicitly in the Introduction, to correct any potential for confusion by readers.
3. There is a difference between the biological significance of men with masculine vs feminine faces, and men with beards vs those who are clean shaven, in that the latter can exhibit the most dramatic changes to reflect a personal grooming choice, contingent upon personality, culture, social setting, etc. You can shave away facial hair, but not a square jawline. This distinction seems to be missing from the Introduction and Discussion. For example, the Discussion opens by saying: "Craniofacial masculinity and beardedness rely on different androgenic processes [...] potentially reflecting difference mechanisms of sexual selection via female choice influencing their expression. To explore this..." - but it's not clear how the manuscript was exploring "androgenic processes of beardedness" (or sexual selection thereon) rather than men's choices to be clean-shaven or bearded (and women's reactions to that choice). Individual differences in the androgenic processes underlying men's beardedness have very little bearing upon whether any individual man has a beard or not, because he can choose to grow it or shave it off. See also Line 196-7, and elsewhere.
4. Line 107-8: what is the evidence that "men with full beards report higher mating success compared with their less masculine peers"? I can't see any such evidence in the references cited. (I think this is part of the problem outlined in 3. above - there's inappropriate conflation of facial masculinity and beardedness throughout the manuscript).
5. Lines 169 - 172: the authors cite a study that finds a positive relationship between disgust for pathogens and attractiveness ratings of beards, and then conclude that it is unknown whether concern around ectoparasites is associated with attractiveness judgements of facial hair. This seems an odd juxtaposition; perhaps better to refer to the value of replicating studies?

6. Line 178: I don't follow the hypothesis that women who desire to have children should have higher preferences for facial masculinity than women who do not desire offspring. Firstly, it's a hard sell arguing that there's been natural selection (is that what you're arguing?) for variation in desire for having vs not having children, given 1) the negative implications of not having children on reproductive success, and 2) having children (prior to modern contraceptives) is contingent upon a desire for relationships and sex, rather than a desire for children directly. Secondly, couldn't you equally say that women who desire children should also have higher preferences for male facial femininity, given its association with parenting investment?

7. I'm afraid I don't follow the second part of the sentence (beginning 'and bearded men') at Lines 244-6.

8. Lines 281-4: Did your stimuli consist of 3 composite male faces, each made up from photos of 3 individual men? That would require $3 \times 3 = 9$ individual men as the basis for the photos. But 9 individual men isn't consistent with the 37 males described at line 255 - could you explain further?

9. The manuscript refers to 40%, 70%, 100%, 130% and 160% masculine throughout (with the exception of a reference to 30% feminised at line 308). The convention that I've seen much more frequently is to refer to the face that you call '100% masculine' as the unmanipulated face. Then, the faces you call 40% and 70% masculine would be described as 30% and 60% feminised, and the faces that you call 130% and 160% masculine would be described as 30% and 60% masculinised. This change would also be helpful in discussing the results, where the manuscript currently says things like "preferences peaked at intermediate levels of masculinity (70%)..." - in fact, the face you describe as '70% masculine' has been feminised relative to the original men's faces in the images, and so we might expect it to be a face that is more feminine than an average man's face; calling it '70% masculine' is somewhat confusing. I would recommend this change throughout, including incorporation into the Results (e.g. line 402-3, "Medium and high levels of masculinity were rated as more attractive than every other level of masculinity" - in fact I think you mean that the unmanipulated and slightly masculinised faces were rated as more attractive - an important distinction!), as well as the Discussion of the existing literature.

10. The Tables are incorrectly referred to throughout the text, such that 'Table 1' is actually Table 2, and so on.

11. I think Figures 3, 4 and 5 could more usefully label the x-axis with 'disgust scale' rather than 'scale'. What are the units in the legend on the right-hand side of Figure 4? You might want to consider whether an alternative choice of colours would be more helpful for colour blindness and black-and-white printing.

12. Lines 494 - 497 - it might be worth noting that women's strongest preferences in previous studies depend in part upon what stimuli are available. For example, heavy stubble has been judged more attractive than full beards in some research, but in the manuscript, the women only judged full beards against clean shaven men.

13. Line 579-81: this is one way to interpret the correlation, but you could equally say that women's preferences for beardedness influenced their choice of partner.

14. Line 586: I'm afraid I don't understand the reference to maternal health - could you expand what is meant there? How is men's beardedness associated with maternal health?

15. Are the raw data available somewhere?

Review form: Reviewer 2

Is the manuscript scientifically sound in its present form?

Yes

Are the interpretations and conclusions justified by the results?

Yes

Is the language acceptable?

No

Do you have any ethical concerns with this paper?

No

Have you any concerns about statistical analyses in this paper?

No

Recommendation?

Major revision is needed (please make suggestions in comments)

Comments to the Author(s)

The authors' present evidence from a large online sample that high disgust for ectoparasites is related to low preference for male sexual dimorphism (beardedness, facial structure), providing converging evidence for work on disgust and human mate preference. Moreover, while they provide evidence that beardedness is attractive to women (on average) across relationship contexts, their data provide little evidence for contextual and individual variation in women's masculinity preference, in contrast to patterns observed in prior work.

The project overall is excellent, the manuscript fits with the remit of the journal and will interest readers. It has given me the opportunity to reflect on replicability within this body of literature and the introduction alone is a good section to direct any student toward, if they study individual differences in mate preferences. I have a few important comments to address in a revision, which would improve the overall quality of the manuscript.

Models.

There may be potential issues with noise in the data from this large and seemingly age-diverse sample of Mturk users (i.e., who vary in age from the typical student samples used in many of the cited studies). This may lend itself to further analyses to refute my point or, if the authors deem this unwarranted, it should be discussed as a potential limitation within the Discussion section. Specifically, I was wondering about the implications that i) participant age, ii) parity, iii) the screening criteria for orientation (i.e. not limited to exclusively heterosexual women) and iv) current partnership status may have for your models, which may be important for some of your individual differences tests (e.g. the null effect of desire to become pregnant). The average age of the sample suggests that many of the participants might have at least one child so there may or may not be variance in pregnancy desire which includes women whose desire at one point was relatively high (and your sample may even include post-menopausal women).

Theory and interpretation/discussion.

Please double check/revise your Discussion section to look at how findings are interpreted across this section. I am specifically referring to Analysis 1. On my reading, you appear to provide evidence for a general 'aversion' toward feminine facial cues in men rather than a preference for masculinity per se - as such, women in the sample may be equally attracted to androgyny or

averageness as they are exaggerated sex typicality (at least when we take beards out of the equation and then think about how they add to the appeal of facial structure in this way, according to my understanding of your data). I think restructuring/editing this section is important to:

- i) address this issue
- ii) help the reader get a clear picture about beards as a cue to 'quality' and a cue to avoid (according to your review) and how your data fit with this (even if it's just a case of re-stating the two parasite theories more clearly before discussing the implications of your beardedness data)
- iii) To discuss your findings, which use a full-beard morph, in relation to findings where some facial hair (stubble) is optimally attractive.

Layout and manuscript structure

I found the results a bit hard to follow (e.g. line 420 onward), particularly when trying to compare preference across different levels of masculinity. I leave this first point at the authors' discretion but I think that Analysis 1 may actually work better in Table form while the individual difference tests may work better-presented in-text as the authors can condense the effects by presenting null effects together after reporting the positive ones (rather than present quite large tables). In general I think that hypotheses could be labelled in the introduction and discussion to make it easier to cross-reference between sections (i.e., Hypothesis #1, 2 etc.).

I leave it at the authors' discretion but although the Introduction was a very interesting read, I found it started to lose focus around line 196. In my view it may benefit from restructuring/editing to cover i) theory, ii) masculine facial structure, iii) beardedness, iv) individual differences, v) rationale/hypotheses, with relevant sentences linking sections together. Given that it is such a great overview of this body of knowledge, it would be great to increase its written impact with some editing.

I thought the manuscript ended a bit abruptly (and some speculation, in my view, around lines 572) and thought could benefit from a short paragraph re-stating the findings/conclusion.

Apologies if I have missed this but I think Marcinkowska and colleagues' 2018 Hormones and Behavior paper is a relevant citation to include in the Introduction.

Minor edits/tense/typos caught on lines 143, 149, 153, 156, 167, 173, 207, 219, 272, 406 (three decimal places), 465, 513 (wording/use of word 'threshold' within the sentence), 518, 528, 543 and 587.

Thank you for the opportunity to review this interesting work.

Decision letter (RSOS-191209.R0)

08-Sep-2019

Dear Dr Dixon,

The editors assigned to your paper ("A multivariate analysis of women's mating strategies and sexual selection on facial masculinity and beardedness in men.") have now received comments from reviewers. We would like you to revise your paper in accordance with the referee and

Associate Editor suggestions which can be found below (not including confidential reports to the Editor). Please note this decision does not guarantee eventual acceptance.

Please submit a copy of your revised paper before 01-Oct-2019. Please note that the revision deadline will expire at 00.00am on this date. If we do not hear from you within this time then it will be assumed that the paper has been withdrawn. In exceptional circumstances, extensions may be possible if agreed with the Editorial Office in advance. We do not allow multiple rounds of revision so we urge you to make every effort to fully address all of the comments at this stage. If deemed necessary by the Editors, your manuscript will be sent back to one or more of the original reviewers for assessment. If the original reviewers are not available, we may invite new reviewers.

- Data accessibility

<http://datadryad.org/submit?journalID=RSOS&manu=RSOS-191209>

- Competing interests

- Authors' contributions

- Acknowledgements

- Funding statement

on behalf of Dr Rosalind Arden (Associate Editor) and Essi Viding (Subject Editor)
openscience@royalsociety.org

Associate Editor's comments (Dr Rosalind Arden):

Associate Editor: 1

Comments to the Author:

Both reviewers find this a worthwhile and interesting study. The 'major' revision status (rather than minor) is more a matter of many small points; it does not indicate serious problems with the manuscript. Both Reviewers have provided constructive comments many of which may strengthen the paper. We hope you will wish to revise and resubmit this ms to RSOS.

Comments to Author:

Reviewers' Comments to Author:

Reviewer: 1

Comments to the Author(s)

The manuscript reports the results of a study investigating the associations between women's preferences for facial hair and for masculinised / feminised faces, along with various measures such as the women's self-reported disgust and desire for pregnancy. The manuscript contains a

lot of interesting data and a helpfully large sample size, although I do have a number of questions, below.

1. The complete summary of the results, as provided in the Abstract, is as follows: “We found no associations between pathogen disgust, self-perceived mate value or reproductive ambition and facial masculinity preferences. However, the association between women’s attractiveness ratings and their feelings of disgust for ectoparasites was negative, providing evidence for ectoparasite avoidance hypothesis”. There is no mention of any of the other results. This creates the misleading impression that against a background of null findings, the one stand-out result was the association between attractiveness ratings and ectoparasite disgust. This is not true: for example, higher moral disgust was also associated with increased preference for bearded faces, which does not help the ectoparasite avoidance hypothesis. The Abstract should be adjusted to point out that the picture is muddier than that currently portrayed. (The authors might also choose to consider whether there are also other interesting results that could be incorporated in the Abstract, space permitting.)

2. As far as I am aware, there is no evidence that men’s facial hair patterns (density, growth) are associated with their testosterone levels. This is probably worth stating explicitly in the Introduction, to correct any potential for confusion by readers.

3. There is a difference between the biological significance of men with masculine vs feminine faces, and men with beards vs those who are clean shaven, in that the latter can exhibit the most dramatic changes to reflect a personal grooming choice, contingent upon personality, culture, social setting, etc. You can shave away facial hair, but not a square jawline. This distinction seems to be missing from the Introduction and Discussion. For example, the Discussion opens by saying: “Craniofacial masculinity and beardedness rely on different androgenic processes [...] potentially reflecting difference mechanisms of sexual selection via female choice influencing their expression. To explore this...” – but it’s not clear how the manuscript was exploring “androgenic processes of beardedness” (or sexual selection thereon) rather than men’s choices to be clean-shaven or bearded (and women’s reactions to that choice). Individual differences in the androgenic processes underlying men’s beardedness have very little bearing upon whether any individual man has a beard or not, because he can choose to grow it or shave it off. See also Line 196-7, and elsewhere.

4. Line 107-8: what is the evidence that “men with full beards report higher mating success compared with their less masculine peers”? I can’t see any such evidence in the references cited. (I think this is part of the problem outlined in 3. above – there’s inappropriate conflation of facial masculinity and beardedness throughout the manuscript).

5. Lines 169 – 172: the authors cite a study that finds a positive relationship between disgust for pathogens and attractiveness ratings of beards, and then conclude that it is unknown whether concern around ectoparasites is associated with attractiveness judgements of facial hair. This seems an odd juxtaposition; perhaps better to refer to the value of replicating studies?

6. Line 178: I don’t follow the hypothesis that women who desire to have children should have higher preferences for facial masculinity than women who do not desire offspring. Firstly, it’s a hard sell arguing that there’s been natural selection (is that what you’re arguing?) for variation in desire for having vs not having children, given 1) the negative implications of not having children on reproductive success, and 2) having children (prior to modern contraceptives) is contingent upon a desire for relationships and sex, rather than a desire for children directly. Secondly, couldn’t you equally say that women who desire children should also have higher preferences for male facial femininity, given its association with parenting investment?

7. I'm afraid I don't follow the second part of the sentence (beginning 'and bearded men') at Lines 244-6.

8. Lines 281-4: Did your stimuli consist of 3 composite male faces, each made up from photos of 3 individual men? That would require $3 \times 3 = 9$ individual men as the basis for the photos. But 9 individual men isn't consistent with the 37 males described at line 255 - could you explain further?

9. The manuscript refers to 40%, 70%, 100%, 130% and 160% masculine throughout (with the exception of a reference to 30% feminised at line 308). The convention that I've seen much more frequently is to refer to the face that you call '100% masculine' as the unmanipulated face. Then, the faces you call 40% and 70% masculine would be described as 30% and 60% feminised, and the faces that you call 130% and 160% masculine would be described as 30% and 60% masculinised. This change would also be helpful in discussing the results, where the manuscript currently says things like "preferences peaked at intermediate levels of masculinity (70%)..." - in fact, the face you describe as '70% masculine' has been feminised relative to the original men's faces in the images, and so we might expect it to be a face that is more feminine than an average man's face; calling it '70% masculine' is somewhat confusing. I would recommend this change throughout, including incorporation into the Results (e.g. line 402-3, "Medium and high levels of masculinity were rated as more attractive than every other level of masculinity" - in fact I think you mean that the unmanipulated and slightly masculinised faces were rated as more attractive - an important distinction!), as well as the Discussion of the existing literature.

10. The Tables are incorrectly referred to throughout the text, such that 'Table 1' is actually Table 2, and so on.

11. I think Figures 3, 4 and 5 could more usefully label the x-axis with 'disgust scale' rather than 'scale'. What are the units in the legend on the right-hand side of Figure 4? You might want to consider whether an alternative choice of colours would be more helpful for colour blindness and black-and-white printing.

12. Lines 494 - 497 - it might be worth noting that women's strongest preferences in previous studies depend in part upon what stimuli are available. For example, heavy stubble has been judged more attractive than full beards in some research, but in the manuscript, the women only judged full beards against clean shaven men.

13. Line 579-81: this is one way to interpret the correlation, but you could equally say that women's preferences for beardedness influenced their choice of partner.

14. Line 586: I'm afraid I don't understand the reference to maternal health - could you expand what is meant there? How is men's beardedness associated with maternal health?

15. Are the raw data available somewhere?

Reviewer: 2

Comments to the Author(s)

The authors' present evidence from a large online sample that high disgust for ectoparasites is related to low preference for male sexual dimorphism (beardedness, facial structure), providing converging evidence for work on disgust and human mate preference. Moreover, while they provide evidence that beardedness is attractive to women (on average) across relationship

contexts, their data provide little evidence for contextual and individual variation in women's masculinity preference, in contrast to patterns observed in prior work.

The project overall is excellent, the manuscript fits with the remit of the journal and will interest readers. It has given me the opportunity to reflect on replicability within this body of literature and the introduction alone is a good section to direct any student toward, if they study individual differences in mate preferences. I have a few important comments to address in a revision, which would improve the overall quality of the manuscript.

Models.

There may be potential issues with noise in the data from this large and seemingly age-diverse sample of Mturk users (i.e., who vary in age from the typical student samples used in many of the cited studies). This may lend itself to further analyses to refute my point or, if the authors deem this unwarranted, it should be discussed as a potential limitation within the Discussion section. Specifically, I was wondering about the implications that i) participant age, ii) parity, iii) the screening criteria for orientation (i.e. not limited to exclusively heterosexual women) and iv) current partnership status may have for your models, which may be important for some of your individual differences tests (e.g. the null effect of desire to become pregnant). The average age of the sample suggests that many of the participants might have at least one child so there may or may not be variance in pregnancy desire which includes women whose desire at one point was relatively high (and your sample may even include post-menopausal women).

Theory and interpretation/discussion.

Please double check/revise your Discussion section to look at how findings are interpreted across this section. I am specifically referring to Analysis 1. On my reading, you appear to provide evidence for a general 'aversion' toward feminine facial cues in men rather than a preference for masculinity per se – as such, women in the sample may be equally attracted to androgyny or averageness as they are exaggerated sex typicality (at least when we take beards out of the equation and then think about how they add to the appeal of facial structure in this way, according to my understanding of your data). I think restructuring/editing this section is important to:

- i) address this issue
- ii) help the reader get a clear picture about beards as a cue to 'quality' and a cue to avoid (according to your review) and how your data fit with this (even if it's just a case of re-stating the two parasite theories more clearly before discussing the implications of your beardedness data)
- iii) To discuss your findings, which use a full-beard morph, in relation to findings where some facial hair (stubble) is optimally attractive.

Layout and manuscript structure

I found the results a bit hard to follow (e.g. line 420 onward), particularly when trying to compare preference across different levels of masculinity. I leave this first point at the authors' discretion but I think that Analysis 1 may actually work better in Table form while the individual difference tests may work better-presented in-text as the authors can condense the effects by presenting null effects together after reporting the positive ones (rather than present quite large tables). In general I think that hypotheses could be labelled in the introduction and discussion to make it easier to cross-reference between sections (i.e., Hypothesis #1, 2 etc.).

I leave it at the authors' discretion but although the Introduction was a very interesting read, I found it started to lose focus around line 196. In my view it may benefit from restructuring/editing to cover i) theory, ii) masculine facial structure, iii) beardedness, iv) individual differences, v) rationale/hypotheses, with relevant sentences linking sections together.

Given that it is such a great overview of this body of knowledge, it would be great to increase its written impact with some editing.

I thought the manuscript ended a bit abruptly (and some speculation, in my view, around lines 572) and thought could benefit from a short paragraph re-stating the findings/conclusion.

Apologies if I have missed this but I think Marcinkowska and colleagues' 2018 Hormones and Behavior paper is a relevant citation to include in the Introduction.

Minor edits/tense/typos caught on lines 143, 149, 153, 156, 167, 173, 207, 219, 272, 406 (three decimal places), 465, 513 (wording/use of word 'threshold' within the sentence), 518, 528, 543 and 587.

Thank you for the opportunity to review this interesting work.

Author's Response to Decision Letter for (RSOS-191209.R0)

See Appendix A.

RSOS-191209.R1 (Revision)

Review form: Reviewer 1

Is the manuscript scientifically sound in its present form?

No

Are the interpretations and conclusions justified by the results?

No

Is the language acceptable?

Yes

Do you have any ethical concerns with this paper?

No

Have you any concerns about statistical analyses in this paper?

Yes

Recommendation?

Major revision is needed (please make suggestions in comments)

Comments to the Author(s)

Thank you for the updated manuscript and the helpful replies. However, I don't believe that all of the issues have been fully resolved, unfortunately. I'm also sorry to say that the clarifications and my recent re-reading have, unfortunately, led me to note a couple of issues that I'm sorry to

say I didn't grasp previously; I realise that it is irritating to raise these now, but better than the alternative.

1. I'm still concerned by the conflation of the biological ability to grow a beard, and the personal choice to wear a beard or not (as raised in point 3. of my original review). This conflation crops up throughout the manuscript. Here's an example. Lines 193 – 201 discuss how the androgenic processes that underlie facial hair development might communicate a specific aspect of quality that corresponds to differential preferences in women. Given that the study asks women to make attractiveness judgements of men with / without beards, we might assume that the study is attempting to tap in to sexual selection for these androgenic processes. However, lines 202 onwards go on to merge into discussing facial hair grooming (clean-shaven vs 10 days of hair growth post shaving vs fully beard), and of course clean-shaven vs full beard is what the stimuli used in the study actually represent. Next, in the same paragraph, the manuscript switches again from discussing facial hair grooming choices to say: "These patterns in preferences have implications with regards how two sexually dimorphic androgen dependent facial traits operate to enhance men's attractiveness". But being able to grow a beard (something that differs between men, and associated with genetic variation), and choosing whether to shave or not (something that differs in part according to cultural practices), are two different things.

I note the authors' response that they "discuss the different androgenic processes that underpin the expression of both sexually dimorphic traits [facial masculinity and beardedness] in order to situate them in their biological context", but unfortunately the focus on sexual selection and the biology of facial hair growth implies that this is what the study is about, whereas the authors say in their response that the study is about "how facial masculinity and beardedness interact to determine women's ratings of men's facial attractiveness". Yet there's more discussion of the androgenic processes associated with the ability to grow a beard than the cultural practices and other variations that might lead some men to wear a beard and others to remove it.

Reviewer 2 suggested, in their first review, a way of restructuring / editing the manuscript to improve its focus, and that advice might help clarify the study focus.

2. I'm still concerned by the conflation of facial masculinity and beardedness (as raised in point 4. of my original review). For instance, the authors say (line 141 – 143) that attractiveness scores for beardedness do not change across the menstrual cycle, implying that this is a reasonable thing to test. But why should beardedness preferences be affected by the menstrual cycle? While menstrual cycle shifts in preferences for masculine men's faces have been interpreted in the context that masculine faces might indicate men who invest less in long-term relationships (see lines 118-127), bearded men "are judged as more attractive for long-term parentally investing relationships" (line 185).

3. Participants aged up to 70 were asked to rate items such as, "I am looking forward to having a baby one day". This does not make sense; clearly, this is a different question when addressed to participants at the beginning of their adult life compared to those beyond their child-bearing years. (Reviewer 2 raised this issue in one of their comments, and I don't think the issue has been resolved in the redraft). According, the 'Desire for Pregnancy subscale' scores currently have limited value. They could probably be used alongside age data however.

4. A sample size of >900 participants means that the study is highly powered. Thus, we need to use caution in interpreting statistically significant findings with small effect size, such as the interaction between moral disgust and facial masculinity ratings. (In this context, at line 442, when the manuscript reports that "participants higher in moral disgust rated facial masculinity as more attractive", we need to ask, 'more attractive than what?' – contrasts are not reported). Statistical significance is not the same as biological significance.

I do have several other more minor questions / comments. These include asking how the 15 men used as stimuli were selected from the 37 men originally photographed; the origin of the '40 female and 40 male European faces'; correction of several typos; the (missing) distinction between 'mating success' and male-male competition in lines 109-110; missing column headings in Table 2; adding a key to the column headings in the supporting data Excel file. However, I won't elaborate on those or others now / yet, as the issues above are more pressing.

I still think there's a lot of value in the dataset and manuscript, and I regret not being more positive here. Usually, I feel much more positive about a manuscript upon second reading following revisions – the topic feels more familiar and digestion means that it's easier to follow the rationale and study. Consequently, I am concerned here where I still have reservations at this point. I hope I'm not being unfair.

Review form: Reviewer 2

Is the manuscript scientifically sound in its present form?

Yes

Are the interpretations and conclusions justified by the results?

Yes

Is the language acceptable?

Yes

Do you have any ethical concerns with this paper?

No

Have you any concerns about statistical analyses in this paper?

No

Recommendation?

Accept with minor revision (please list in comments)

Comments to the Author(s)

I thank the authors for their careful and detailed response to my comments. The findings on beardedness and reproductive ambition are definitely noteworthy, in my view, when incorporating partnership status into the model. I see no further issues to address in the manuscript except for minor typos caught when reviewing the edited lines highlighted in their response document:

Line 457-459 – you say 'relationship status' twice when I think you mean to say 'facial masculinity preference'.

Line 481 'when re-running the analysis...' and lines 487 'partnered and single women' and 489 'the amount of facial hair'.

Thanks again for the opportunity to review the work.

Decision letter (RSOS-191209.R1)

04-Nov-2019

Dear Dr Dixon:

On behalf of the Editors, I am pleased to inform you that your Manuscript RSOS-191209.R1 entitled "A multivariate analysis of individual differences in women's mating strategies and sexual selection on men's facial masculinity and beardedness." has been accepted for publication in Royal Society Open Science subject to minor revision in accordance with the referee suggestions. Please find the referees' comments at the end of this email.

The reviewers and Subject Editor have recommended publication, but also suggest some minor revisions to your manuscript. Therefore, I invite you to respond to the comments and revise your manuscript.

- Ethics statement

- Data accessibility

<http://datadryad.org/submit?journalID=RSOS&manu=RSOS-191209.R1>

- Competing interests

- Authors' contributions

- Acknowledgements

- Funding statement

Because the schedule for publication is very tight, it is a condition of publication that you submit the revised version of your manuscript before 13-Nov-2019. Please note that the revision deadline will expire at 00.00am on this date. If you do not think you will be able to meet this date please let me know immediately.

Supplementary files will be published alongside the paper on the journal website and posted on

the online figshare repository (<https://figshare.com>). The heading and legend provided for each supplementary file during the submission process will be used to create the figshare page, so please ensure these are accurate and informative so that your files can be found in searches. Files on figshare will be made available approximately one week before the accompanying article so that the supplementary material can be attributed a unique DOI.

Kind regards,
 Andrew Dunn
 Senior Publishing Editor
 Royal Society Open Science Editorial Office
 Royal Society Open Science
openscience@royalsociety.org

on behalf of Dr Rosalind Arden (Associate Editor) and Essi Viding (Subject Editor)
openscience@royalsociety.org

Associate Editor Comments to Author (Dr Rosalind Arden)(post-review):
 Thank you for your revised ms. The paper is interesting; the data are well worth writing up. Reviewer 2 notes a few typos. Reviewer 1, who also kindly re-read the ms, still has some concerns. Would you be willing to read these comments and respond to them? Reviewers were not of one mind concerning major/minor revisions. I have chosen minor since I hope they will not take too much time and to show that we will warmly welcome a revision.

Associate Editor Comments to the Author (pre-review):
 Thank you for this revision and for your great responses to the Reviewers' comments. I have some additional thoughts:

You didn't measure political conservatism so it seems going beyond evidence in this paper to mention it in the abstract. It seems especially odd given that some very left leaning sub-cultures sport beards as a current fashion trend.

Age. Is there evidence of a shift in fashion for beards by age of the women who participated?

There are lots of places in the ms that are hard to read. Instead of phrases like 'women's preference for beardedness was negatively associated with' could you say women who liked beards/beardedness more also found ectoparasites more disgusting - or whatever you did find. There are lots of sentences that could be written more clearly, also some that do not parse.

Line 146/7 does not make sense because you can't have a pref for dimorphism - dimorphism applies to a group not an individual.

There are some typos (inevitably!). A careful re-read should dislodge grammatical or punctuation blips.

Lines 240, 479, 603, 615 were some I picked up.

Descriptive Statistics. It's helpful for readers to see a table that sets out the n, mean and range of all the measures. You have graphs where the y axis does not start at zero. It's hard for readers to

know if the results are a vanity of small differences without knowing the scale and what the range of values was among the measures.

Minor work would enhance this interesting piece of work, and it would be super to have the Reviewers own thoughts on the revision.

Reviewer comments to Author:

Reviewer: 2

Comments to the Author(s)

I thank the authors for their careful and detailed response to my comments. The findings on beardedness and reproductive ambition are definitely noteworthy, in my view, when incorporating partnership status into the model. I see no further issues to address in the manuscript except for minor typos caught when reviewing the edited lines highlighted in their response document:

Line 457-459 – you say ‘relationship status’ twice when I think you mean to say ‘facial masculinity preference’.

Line 481 ‘when re-running the analysis...’ and lines 487 ‘partnered and single women’ and 489 ‘the amount of facial hair’.

Thanks again for the opportunity to review the work.

Reviewer: 1

Comments to the Author(s)

Thank you for the updated manuscript and the helpful replies. However, I don’t believe that all of the issues have been fully resolved, unfortunately. I’m also sorry to say that the clarifications and my recent re-reading have, unfortunately, led me to note a couple of issues that I’m sorry to say I didn’t grasp previously; I realise that it is irritating to raise these now, but better than the alternative.

1. I’m still concerned by the conflation of the biological ability to grow a beard, and the personal choice to wear a beard or not (as raised in point 3. of my original review). This conflation crops up throughout the manuscript. Here’s an example. Lines 193 – 201 discuss how the androgenic processes that underlie facial hair development might communicate a specific aspect of quality that corresponds to differential preferences in women. Given that the study asks women to make attractiveness judgements of men with / without beards, we might assume that the study is attempting to tap in to sexual selection for these androgenic processes. However, lines 202 onwards go on to merge into discussing facial hair grooming (clean-shaven vs 10 days of hair growth post shaving vs fully beard), and of course clean-shaven vs full beard is what the stimuli used in the study actually represent. Next, in the same paragraph, the manuscript switches again from discussing facial hair grooming choices to say: “These patterns in preferences have implications with regards how two sexually dimorphic androgen dependent facial traits operate to enhance men’s attractiveness”. But being able to grow a beard (something that differs between men, and associated with genetic variation), and choosing whether to shave or not (something that differs in part according to cultural practices), are two different things.

I note the authors’ response that they “discuss the different androgenic processes that underpin the expression of both sexually dimorphic traits [facial masculinity and beardedness] in order to situate them in their biological context”, but unfortunately the focus on sexual selection and the

biology of facial hair growth implies that this is what the study is about, whereas the authors say in their response that the study is about “how facial masculinity and beardedness interact to determine women’s ratings of men’s facial attractiveness”. Yet there’s more discussion of the androgenic processes associated with the ability to grow a beard than the cultural practices and other variations that might lead some men to wear a beard and others to remove it.

Reviewer 2 suggested, in their first review, a way of restructuring / editing the manuscript to improve its focus, and that advice might help clarify the study focus.

2. I’m still concerned by the conflation of facial masculinity and beardedness (as raised in point 4. of my original review). For instance, the authors say (line 141 – 143) that attractiveness scores for beardedness do not change across the menstrual cycle, implying that this is a reasonable thing to test. But why should beardedness preferences be affected by the menstrual cycle? While menstrual cycle shifts in preferences for masculine men’s faces have been interpreted in the context that masculine faces might indicate men who invest less in long-term relationships (see lines 118-127), bearded men “are judged as more attractive for long-term parentally investing relationships” (line 185).

3. Participants aged up to 70 were asked to rate items such as, “I am looking forward to having a baby one day”. This does not make sense; clearly, this is a different question when addressed to participants at the beginning of their adult life compared to those beyond their child-bearing years. (Reviewer 2 raised this issue in one of their comments, and I don’t think the issue has been resolved in the redraft). According, the ‘Desire for Pregnancy subscale’ scores currently have limited value. They could probably be used alongside age data however.

4. A sample size of >900 participants means that the study is highly powered. Thus, we need to use caution in interpreting statistically significant findings with small effect size, such as the interaction between moral disgust and facial masculinity ratings. (In this context, at line 442, when the manuscript reports that “participants higher in moral disgust rated facial masculinity as more attractive”, we need to ask, ‘more attractive than what?’ – contrasts are not reported). Statistical significance is not the same as biological significance.

I do have several other more minor questions / comments. These include asking how the 15 men used as stimuli were selected from the 37 men originally photographed; the origin of the ‘40 female and 40 male European faces’; correction of several typos; the (missing) distinction between ‘mating success’ and male-male competition in lines 109-110; missing column headings in Table 2; adding a key to the column headings in the supporting data Excel file. However, I won’t elaborate on those or others now / yet, as the issues above are more pressing.

I still think there’s a lot of value in the dataset and manuscript, and I regret not being more positive here. Usually, I feel much more positive about a manuscript upon second reading following revisions – the topic feels more familiar and digestion means that it’s easier to follow the rationale and study. Consequently, I am concerned here where I still have reservations at this point. I hope I’m not being unfair.

Author's Response to Decision Letter for (RSOS-191209.R1)

See Appendix B.

Decision letter (RSOS-191209.R2)

26-Nov-2019

Dear Dr Dixon,

It is a pleasure to accept your manuscript entitled "A multivariate analysis of women's mating strategies and sexual selection on men's facial morphology." in its current form for publication in Royal Society Open Science. The comments of the reviewer(s) who reviewed your manuscript are included at the foot of this letter.

Kind regards,
Andrew Dunn
Royal Society Open Science
openscience@royalsociety.org

on behalf of Dr Rosalind Arden (Associate Editor) and Essi Viding (Subject Editor)
openscience@royalsociety.org

Appendix A

Dear Dr. Arden,

We are very grateful for the thoughtful comments from two anonymous Reviewers on our manuscript “A multivariate analysis of women’s mating strategies and sexual selection on facial masculinity and beardedness in men (RSOS-191209)”. We are also grateful for the opportunity to address these comments in a resubmission to Royal Society Open Science.

We have numbered each comment or question from the Reviewers and provided a response immediately below. We have endeavored to make every change and respond to each comment in as much detail as possible. Once again, we want to thank the Reviewers for taking the time to critique and Review our manuscript, which we believe it is significantly improved from the first submission and we hope it is now in a state that is more appropriate for publication.

Associate Editor

Both reviewers find this a worthwhile and interesting study. The 'major' revision status (rather than minor) is more a matter of many small points; it does not indicate serious problems with the manuscript. Both Reviewers have provided constructive comments many of which may strengthen the paper. We hope you will wish to revise and resubmit this ms to RSOS.

Response: We are grateful for opportunity to revise our paper for Royal Society Open Science and we agree that the comments are minor and we have addressed each one below in detail. We hope the Reviewers are satisfied with our responses.

Reviewer: 1

Comments to the Author(s)

The manuscript reports the results of a study investigating the associations between women’s preferences for facial hair and for masculinised / feminised faces, along with various measures such as the women’s self-reported disgust and desire for pregnancy. The manuscript contains a lot of interesting data and a helpfully large sample size, although I do have a number of questions, below.

Response: We thank the Reviewer and we have numbered each question and provided our response immediately below.

1. The complete summary of the results, as provided in the Abstract, is as follows: “We found no associations between pathogen disgust, self-perceived mate value or reproductive ambition and facial masculinity preferences. However, the association between women’s attractiveness ratings and their feelings of disgust for ectoparasites was negative, providing evidence for ectoparasite avoidance hypothesis”. There is no mention of any of the other results. This creates the misleading impression that against a background of null findings, the one stand-out result was

the association between attractiveness ratings and ectoparasite disgust. This is not true: for example, higher moral disgust was also associated with increased preference for bearded faces, which does not help the ectoparasite avoidance hypothesis. The Abstract should be adjusted to point out that the picture is muddier than that currently portrayed. (The authors might also choose to consider whether there are also other interesting results that could be incorporated in the Abstract, space permitting.)

Response: We agree with the Reviewer and we have rewritten the abstract to more fully capture the complexities of our results, including the moral disgust associations and those from the reproductive ambition analyses, in order to highlight that our multivariate approach reveals a more nuanced set of associations than is reported in the previous literature.

2. As far as I am aware, there is no evidence that men's facial hair patterns (density, growth) are associated with their testosterone levels. This is probably worth stating explicitly in the Introduction, to correct any potential for confusion by readers.

Response: The Reviewer is correct and this is now stated in the introduction on Lines 91-94:

“Although facial masculinity and facial hair both require testosterone for their development and full expression in adulthood, total testosterone levels alone do not explain variation in the pattern, density and distribution of beardedness in men. Instead, beards develop as testosterone is synthesised into dihydrotestosterone via 5-alpha reductase activity within hair follicles (22,23).”

3. There is a difference between the biological significance of men with masculine vs feminine faces, and men with beards vs those who are clean shaven, in that the latter can exhibit the most dramatic changes to reflect a personal grooming choice, contingent upon personality, culture, social setting, etc. You can shave away facial hair, but not a square jawline. This distinction seems to be missing from the Introduction and Discussion. For example, the Discussion opens by saying: “Craniofacial masculinity and beardedness rely on different androgenic processes [...] potentially reflecting difference mechanisms of sexual selection via female choice influencing their expression. To explore this...” – but it's not clear how the manuscript was exploring “androgenic processes of beardedness” (or sexual selection thereon) rather than men's choices to be clean-shaven or bearded (and women's reactions to that choice). Individual differences in the androgenic processes underlying men's beardedness have very little bearing upon whether any individual man has a beard or not, because he can choose to grow it or shave it off. See also Line 196-7, and elsewhere.

Response: The Reviewer is correct that this part of the introduction did not accurately reflect what our experimental manipulations were testing. We discuss the different androgenic processes that underpin the expression of both sexually dimorphic traits to situate them in their biological context. However, the Reviewer is absolutely correct that facial hair can be groomed or removed entirely, whereas craniofacial traits cannot. We have rewritten the manuscript to state that we were testing how facial masculinity and beardedness interact to determine women's ratings of

men's facial attractiveness and to highlight that grooming a key factor underlying displays of masculinity in men (Lines 193-201 of the introduction and Lines 499-506 of the Discussion).

4. Line 107-8: what is the evidence that “men with full beards report higher mating success compared with their less masculine peers”? I can't see any such evidence in the references cited. (I think this is part of the problem outlined in 3. above – there's inappropriate conflation of facial masculinity and beardedness throughout the manuscript).

Response: We apologise for not being clearer in this part of the introduction. The evidence that the choice to wear facial hair is associated with mating success comes from several studies that we cited in this section. Firstly, Barber (2001) used time series data on male facial hair fashions scored from photographs from men announcing their marriages in the London from 1842-1972 and found that more men were bearded in times when the marriage market was more male-biased. This suggests that masculinity is amplified when intra-sexual competition is strongest and is associated with higher mating success. More recently, Dixson et al (2017; 2019) reported cross-cultural data showing that beards were more common in larger cities where average incomes were low income but health indices were high, sex ratios are more male biased and female preferences for facial hair are strongest. This again suggests an association between women's preferences for beards under conditions where male-male competition is stronger. We have rewritten this part of the manuscript to better explain how facial hair is associated with men's attractiveness and mating success (Lines 107-113).

5. Lines 169 – 172: the authors cite a study that finds a positive relationship between disgust for pathogens and attractiveness ratings of beards, and then conclude that it is unknown whether concern around ectoparasites is associated with attractiveness judgements of facial hair. This seems an odd juxtaposition; perhaps better to refer to the value of replicating studies?

Response: We agree with the Reviewer and have rewritten this part of the manuscript to state that we are replicating a study reporting a positive association between preferences for men's beards and pathogen disgust (Lines 169-173).

6. Line 178: I don't follow the hypothesis that women who desire to have children should have higher preferences for facial masculinity than women who do not desire offspring. Firstly, it's a hard sell arguing that there's been natural selection (is that what you're arguing?) for variation in desire for having vs not having children, given 1) the negative implications of not having children on reproductive success, and 2) having children (prior to modern contraceptives) is contingent upon a desire for relationships and sex, rather than a desire for children directly. Secondly, couldn't you equally say that women who desire children should also have higher preferences for male facial femininity, given its association with parenting investment?

Response: We understand the Reviewer's confusion regarding the hypothesis that women desiring children should prefer more masculine looking men. The hypothesis was generated by Watkins (2012), who reported a positive association between women's desire for pregnancy and

facial masculinity among women in relationships but not among single women. Watkins (2012) used mating strategies theory to predict that as mate preferences for male long-term health among women may be stronger among women who are more likely to conceive, preferences for masculine facial shape cues (which are associated with some aspects of health) would be stronger among women with greater reproductive ambition. We agree with the Reviewer that this hypothesis is a hard sell, especially as the literature on variation in women's mate preferences due to changes in fertility over the menstrual has not been replicated in several studies. However, we wanted to include this variable in our multivariate study as the findings have never been replicated. Thus, we have rewritten this hypothesis to better explain our rationale for its inclusion in our study and that we were seeking to replicate a previously published effect (Lines 179-184).

7. I'm afraid I don't follow the second part of the sentence (beginning 'and bearded men') at Lines 244-6.

Response: We apologise that this sentence was poorly written. In our resubmission, we have rewritten it on lines 250-253 to read:

“Most recently, mothers reported stronger preferences for beardedness when judging parenting skills but not attractiveness compared to women without children (49) and women in long-term relationships with bearded men reported higher reproductive success than women in long-term relationships with non-bearded men”.

We hope this is now easier to understand.

8. Lines 281-4: Did your stimuli consist of 3 composite male faces, each made up from photos of 3 individual men? That would require $3 \times 3 = 9$ individual men as the basis for the photos. But 9 individual men isn't consistent with the 37 males described at line 255 - could you explain further?

Response: Our stimuli consisted of three composite clean-shaven and when bearded faces. Each composite face was made up of the same 5 men photographed when clean-shaven and fully bearded. For each of the composites, the 5 men were drawn at random from the full library of 37 male faces each of whom were photographed when clean-shaven and with full beards. We have rewritten the methods (see Lines 267-279) and we hope this is now clearer to follow.

9. The manuscript refers to 40%, 70%, 100%, 130% and 160% masculine throughout (with the exception of a reference to 30% feminised at line 308). The convention that I've seen much more frequently is to refer to the face that you call '100% masculine' as the unmanipulated face. Then, the faces you call 40% and 70% masculine would be described as 30% and 60% feminised, and the faces that you call 130% and 160% masculine would be described as 30% and 60% masculinised. This change would also be helpful in discussing the results, where the manuscript currently says things like “preferences peaked at intermediate levels of masculinity (70%)...” – in fact, the face you describe as '70% masculine' has been feminised relative to the original

men's faces in the images, and so we might expect it to be a face that is more feminine than an average man's face; calling it '70% masculine' is somewhat confusing. I would recommend this change throughout, including incorporation into the Results (e.g. line 402-3, "Medium and high levels of masculinity were rated as more attractive than every other level of masculinity" – in fact I think you mean that the unmanipulated and slightly masculinised faces were rated as more attractive – an important distinction!), as well as the Discussion of the existing literature.

Response: This is a very helpful suggestion and we have incorporated it into our revised manuscript, including the Figures, Introduction, Results and Discussion. We hope this has made interpreting the manuscript easier.

10. The Tables are incorrectly referred to throughout the text, such that 'Table 1' is actually Table 2, and so on.

Response: We are very grateful to the Reviewer for picking up on this oversight. We have now corrected the numbering for the tables throughout the results section.

11. I think Figures 3, 4 and 5 could more usefully label the x-axis with 'disgust scale' rather than 'scale'. What are the units in the legend on the right-hand side of Figure 4? You might want to consider whether an alternative choice of colours would be more helpful for colour blindness and black-and-white printing.

Response: We agree with the Reviewer and we have changed the label on the x-axis of the figures 3, 4 and 5 to read 'disgust scale'. Additionally, as suggested and we have altered the colours in the regression lines and the legend to Figure 4. We hope they are now more appropriate.

12. Lines 494 – 497 – it might be worth noting that women's strongest preferences in previous studies depend in part upon what stimuli are available. For example, heavy stubble has been judged more attractive than full beards in some research, but in the manuscript, the women only judged full beards against clean shaven men.

Response: We agree with the Reviewer and we have rewritten this part of the discussion on lines 507-512 to read:

"Past research has reported largely equivocal preferences for beardedness among women (39), partly due to the degree of facial hair presented in the stimuli whereby light facial hair or 'stubble' is more attractive than full beards and clean-shaven faces in some studies (23,25,26,109). However, our finding that women judged beards as more attractive for long-term than short-term relationships supports several previous studies (23,25,26,39,49)."

We hope this captures the Reviewer's suggestions sufficiently.

13. Line 579-81: this is one way to interpret the correlation, but you could equally say that women's preferences for beardedness influenced their choice of partner.

Response: The Reviewer raises another important suggestion and we have adjusted the text on Lines 589-593 to read:

“While we found no associations between women's attractiveness ratings for beards and beardedness in their father's, women in relationships with bearded men had stronger preferences for bearded partners. This may reflect that social exposure to beardedness influences the strength of women's preferences for facial hair (135) or that preferences for beardedness has an influence on women's choice of beardedness in their actual partners.”

We thank the Reviewer for this suggestion.

14. Line 586: I'm afraid I don't understand the reference to maternal health – could you expand what is meant there? How is men's beardedness associated with maternal health?

Response: We apologise and agree that this is confusing. We have deleted this statement from the revised text.

15. Are the raw data available somewhere?

Response: Yes, the data are available on DRYAD via the link below.

<https://datadryad.org/review?doi=doi:10.5061/dryad.t9d333h>

Reviewer: 2

Comments to the Author(s)

The authors' present evidence from a large online sample that high disgust for ectoparasites is related to low preference for male sexual dimorphism (beardedness, facial structure), providing converging evidence for work on disgust and human mate preference. Moreover, while they provide evidence that beardedness is attractive to women (on average) across relationship contexts, their data provide little evidence for contextual and individual variation in women's masculinity preference, in contrast to patterns observed in prior work.

The project overall is excellent, the manuscript fits with the remit of the journal and will interest readers. It has given me the opportunity to reflect on replicability within this body of literature and the introduction alone is a good section to direct any student toward, if they study individual differences in mate preferences. I have a few important comments to address in a revision, which would improve the overall quality of the manuscript.

Response: We are grateful for the Reviewer's positive comments and we hope we have satisfied their concerns in our responses below.

1. Models.

There may be potential issues with noise in the data from this large and seemingly age-diverse sample of Mturk users (i.e., who vary in age from the typical student samples used in many of the cited studies). This may lend itself to further analyses to refute my point or, if the authors deem this unwarranted, it should be discussed as a potential limitation within the Discussion section. Specifically, I was wondering about the implications that i) participant age, ii) parity, iii) the screening criteria for orientation (i.e. not limited to exclusively heterosexual women) and iv) current partnership status may have for your models, which may be important for some of your individual differences tests (e.g. the null effect of desire to become pregnant). The average age of the sample suggests that many of the participants might have at least one child so there may or may not be variance in pregnancy desire which includes women whose desire at one point was relatively high (and your sample may even include post-menopausal women).

Response: In planning our statistical analyses we were cautious not to over-fit our models and thereby risk generating spurious effects. However, we agree with the Reviewer that some of these factors are worth investigating in our analyses and we have re-analysed the reproductive ambition data to include women's current relationship status. Indeed, the original findings from Watkins (2012) reported significant associations between facial masculinity preferences among women currently in relationships, but not among single women. In our analyses, we found no associations between reproductive ambition, relationship status and preferences for facial masculinity. However, we found that among women in relationships and single women preferences for beardedness were positively associated with reproductive ambition, while preferences for clean-shaven faces were positively associated with reproductive ambition among single women but negatively with preferences among women in long-term relationships. We have included this analyses as it relates to our hypotheses in the introduction (Lines 226-232), we have reported these findings in the results (Lines 453-458 and Lines 480-487), included a new Figure 6 showing the results relating to preferences for beardedness and incorporated the results into the discussion discussion (Lines 556-558 and Lines 597-605) of the revised manuscript. With regards using age as a proxy for parity and reproductive capabilities, we have elected not to run those analyses as we did not directly measure parity or reproductive status and therefore cannot conclude that our participants had children or were had undergone menopause. However, we have noted these as limitations to our study and potential areas for future studies in the revised final paragraph of the discussion (Lines 606-628).

2. Theory and interpretation/discussion.

Please double check/revise your Discussion section to look at how findings are interpreted across this section. I am specifically referring to Analysis 1. On my reading, you appear to provide evidence for a general 'aversion' toward feminine facial cues in men rather than a preference for masculinity per se – as such, women in the sample may be equally attracted to androgyny or averageness as they are exaggerated sex typicality (at least when we take beards out of the

equation and then think about how they add to the appeal of facial structure in this way, according to my understanding of your data). I think restructuring/editing this section is important to:

i) address this issue

Response: We have revised the discussion to highlight a potential role for androgyny enhancing male attractiveness in this sample and a general avoidance of overly feminine looking faces in clean-shaven faces (Lines 518-527).

ii) help the reader get a clear picture about beards as a cue to 'quality' and a cue to avoid (according to your review) and how your data fit with this (even if it's just a case of re-stating the two parasite theories more clearly before discussing the implications of your beardedness data)

Response: We agree with the Reviewer's suggestion and we have restated how parasite-stress models and ectoparasite-avoidance models of sexual selection are thought to explain variation in female preferences for male beardedness before discussing the findings of the current study (Lines 528-534).

iii) To discuss your findings, which use a full-beard morph, in relation to findings where some facial hair (stubble) is optimally attractive. -

Response: We agree with the Reviewer and we have discussed this issue in the revised manuscript (Lines 508-511).

3. Layout and manuscript structure

I found the results a bit hard to follow (e.g. line 420 onward), particularly when trying to compare preference across different levels of masculinity. I leave this first point at the authors' discretion but I think that Analysis 1 may actually work better in Table form while the individual difference tests may work better-presented in-text as the authors can condense the effects by presenting null effects together after reporting the positive ones (rather than present quite large tables). In general I think that hypotheses could be labelled in the introduction and discussion to make it easier to cross-reference between sections (i.e., Hypothesis #1, 2 etc.).

Response: We understand that the way the results section for Analysis 1 is currently written is difficult to follow. We have chosen to retain the general formatting, but we have rewritten parts of the introduction to better capture the direction of our findings and reduce its length. We hope it is now easier to understand. We have also decided to retain the Tables as they include the full findings. However, we have followed the Reviewer's suggestions to label our hypotheses in the introduction and referred to them in the results and discussion sections. We hope this has improved the readability of our results in relation to the introduction and discussion.

4. I leave it at the authors' discretion but although the Introduction was a very interesting read, I found it started to lose focus around line 196. In my view, it may benefit from restructuring/editing to cover i) theory, ii) masculine facial structure, iii) beardedness, iv) individual differences, v) rationale/hypotheses, with relevant sentences linking sections together. Given that it is such a great overview of this body of knowledge, it would be great to increase its written impact with some editing.

Response: Following the advice given in this comment, we have edited the introduction to improve the flow between sections. We hope this has improved the overall readability of the introduction leading into the hypotheses.

5. I thought the manuscript ended a bit abruptly (and some speculation, in my view, around lines 572) and thought could benefit from a short paragraph re-stating the findings/conclusion.

Response: We agree with the Reviewer and we have added some limitations to our study and restated some of the main findings regarding social aspects of conservatism and pathogens underpinning women's preferences for men's beardedness in a new final paragraph to the discussion (607-629).

6. Apologies if I have missed this but I think Marcinkowska and colleagues' 2018 Hormones and Behavior paper is a relevant citation to include in the Introduction.

Response: We have cited Marcinkowska et al., 2018 Psychoneuroendocrinology rather than the Hormones and Behavior paper which reports on preferences for masculinity in male faces when accounting for with estrogen/progesterone ratios and current relationship status.

7. Minor edits/tense/typos caught on lines 143, 149, 153, 156, 167, 173, 207, 219, 272, 406 (three decimal places), 465, 513 (wording/use of word 'threshold' within the sentence), 518, 528, 543 and 587.

Response: We are grateful to the Reviewer for identifying these errors in the text and we have corrected them in the revised manuscript.

8. Thank you for the opportunity to review this interesting work.

Response: We thank the Reviewer for their helpful comments and we hope that we have answered their questions in sufficient detail.

Appendix B

Dear Dr. Arden,

We are very pleased that our manuscript “A multivariate analysis of women’s mating strategies and sexual selection on men’s facial morphology (RSOS-191209)” has been accepted for publication pending minor revisions at Royal Society Open Science. We have numbered each comment or question from the Reviewers and provided a response immediately below. We done our best to make every change and respond to each comment in as much detail as possible. We hope our paper is now appropriate for publication.

Associate Editor Comments to Author (Dr Rosalind Arden)(post-review):

Thank you for your revised ms. The paper is interesting; the data are well worth writing up. Reviewer 2 notes a few typos. Reviewer 1, who also kindly re-read the ms, still has some concerns. Would you be willing to read these comments and respond to them? Reviewers were not of one mind concerning major/minor revisions. I have chosen minor since I hope they will not take too much time and to show that we will warmly welcome a revision.

Associate Editor Comments to the Author (pre-review):

Thank you for this revision and for your great responses to the Reviewers' comments. I have some additional thoughts:

1. You didn't measure political conservatism so it seems going beyond evidence in this paper to mention it in the abstract. It seems especially odd given that some very left leaning sub-cultures sport beards as a current fashion trend.

Response: We agree and we have deleted these statements from the abstract.

2. Age. Is there evidence of a shift in fashion for beards by age of the women who participated?

Response: There was not a significant association between preferences for beards and participant’s age. We have reported this in the revised results (Line 512-516).

3. There are lots of places in the ms that are hard to read. Instead of phrases like 'women's preference for beardedness was negatively associated with' could you say women who liked beards/beardedness more also found ectoparasites more disgusting - or whatever you did find. There are lots of sentences that could be written more clearly, also some that do not parse.

Response: We understand that the paper is difficult to read at points and we have rewritten the parts where we sound overly repetitive. We hope the paper is easier to follow now.

4. Line 146/7 does not make sense because you can't have a pref for dimorphism - dimorphism applies to a group not an individual.

Response: We agree and we have changed the text (Lines 149-150).

5. There are some typos (inevitably!). A careful re-read should dislodge grammatical or punctuation blips. Lines 240, 479, 603, 615 were some I picked up.

Response: Thank you we have corrected these typos.

6. Descriptive Statistics. It's helpful for readers to see a table that sets out the n, mean and range of all the measures. You have graphs where the y axis does not start at zero. It's hard for readers to know if the results are a vanity of small differences without knowing the scale and what the range of values was among the measures.

Response: We have included a new table of descriptive statistics for the surveys (Table 1) and we have noted the lower and upper bounds of the rating scales in the Figure captions.

Reviewer comments to Author:

Reviewer: 2

Comments to the Author(s)

I thank the authors for their careful and detailed response to my comments. The findings on beardedness and reproductive ambition are definitely noteworthy, in my view, when incorporating partnership status into the model. I see no further issues to address in the manuscript except for minor typos caught when reviewing the edited lines highlighted in their response document:

1. Line 457-459 – you say ‘relationship status’ twice when I think you mean to say ‘facial masculinity preference’.

Response: Thank you, we have adjusted the text accordingly.

2. Line 481 ‘when re-running the analysis...’ and lines 487 ‘partnered and single women’ and 489 ‘the amount of facial hair’.

Response: Thank you, we have corrected these parts of the text.

Thanks again for the opportunity to review the work.

Reviewer: 1

Comments to the Author(s)

Thank you for the updated manuscript and the helpful replies. However, I don’t believe that all of the issues have been fully resolved, unfortunately. I’m also sorry to say that the clarifications and my recent re-reading have, unfortunately, led me to note a couple of issues that I’m sorry to say I didn’t grasp previously; I realise that it is irritating to raise these now, but better than the alternative.

1. I’m still concerned by the conflation of the biological ability to grow a beard, and the personal choice to wear a beard or not (as raised in point 3. of my original review). This conflation crops up throughout the manuscript. Here’s an example. Lines 193 – 201 discuss how the androgenic processes that underlie facial hair development might communicate a specific aspect of quality that corresponds to differential preferences in women. Given that the study asks women to make attractiveness judgements of men with / without beards, we might assume that the study is attempting to tap in to sexual selection for these androgenic processes. However, lines 202 onwards go on to merge into discussing facial hair grooming (clean-shaven vs 10 days of hair growth post shaving vs fully beard), and of course clean-shaven vs full beard is what the stimuli used in the study actually represent. Next, in the same paragraph, the manuscript switches again from discussing facial hair grooming choices to say: “These patterns in preferences have implications with regards how two sexually dimorphic androgen dependent facial traits operate to enhance men’s attractiveness”. But being able to grow a beard (something that differs between men, and associated with genetic variation), and choosing whether to shave or not (something that differs in part according to cultural practices), are two different things.

I note the authors' response that they "discuss the different androgenic processes that underpin the expression of both sexually dimorphic traits [facial masculinity and beardedness] in order to situate them in their biological context", but unfortunately the focus on sexual selection and the biology of facial hair growth implies that this is what the study is about, whereas the authors say in their response that the study is about "how facial masculinity and beardedness interact to determine women's ratings of men's facial attractiveness". Yet there's more discussion of the androgenic processes associated with the ability to grow a beard than the cultural practices and other variations that might lead some men to wear a beard and others to remove it.

Response: We thank the Reviewer for their comment. We were outlining the biological differences in androgenic processes that underpin the development of facial hair and craniofacial masculinity as it generates different predictions regarding women's mate preferences. However, we agree with the Reviewer that a better discussion of the cultural effects of grooming on women's attractiveness judgments of men's facial hair is warranted. We have rewritten a paragraph in the discussion that gives more background on how cultural patterns in grooming are associated with men's decisions to wear full beards and women's attractiveness judgments of facial hair (Lines 615-626).

2. I'm still concerned by the conflation of facial masculinity and beardedness (as raised in point 4. of my original review). For instance, the authors say (line 141 – 143) that attractiveness scores for beardedness do not change across the menstrual cycle, implying that this is a reasonable thing to test. But why should beardedness preferences be affected by the menstrual cycle? While menstrual cycle shifts in preferences for masculine men's faces have been interpreted in the context that masculine faces might indicate men who invest less in long-term relationships (see lines 118-127), bearded men "are judged as more attractive for long-term parentally investing relationships" (line 185).

Response: We understand the Reviewer's concerns. We were summarising the literature on menstrual cycle effects on women preferences for facial masculinity and beardedness because, in its simplest iteration, the ovulatory shift hypothesis suggests that masculine traits are more attractive to women when they are at the fertile phase of the menstrual cycle. However, subsequent iterations of this hypothesis are more complex and suggest that mating strategies might underpin menstrual cycle shifts, so that preferences would be strongest for masculine traits at when judging short-term rather than long-term relationships. This hypothesis is not supported for beards, and the Reviewer is correct that this may be because beards are preferred for long-term and paternally investing relationships. We have changed the text to make this clearer (Lines 144-148).

3. Participants aged up to 70 were asked to rate items such as, "I am looking forward to having a baby one day". This does not make sense; clearly, this is a different question when addressed to participants at the beginning of their adult life compared to those beyond their child-bearing years. (Reviewer 2 raised this issue in one of their comments, and I don't think the issue has been resolved in the redraft). According, the 'Desire for Pregnancy subscale' scores currently have limited value. They could probably be used alongside age data however.

Response: We agree with the Reviewer and we have run our analyses again including participant's age. This analysis found no associations between age, preferences for facial masculinity or beardedness and reproductive ambition. We have stated this finding in the results section (Lines 512-516) and we have included the analyses in an R markdown in the electronic supplementary materials (ESM 2).

4. A sample size of >900 participants means that the study is highly powered. Thus, we need to use caution in interpreting statistically significant findings with small effect size, such as the interaction between moral disgust and facial masculinity ratings. (In this context, at line 442, when the manuscript reports that "participants higher in moral disgust rated facial masculinity as more attractive", we need to ask, 'more attractive than what?' – contrasts are not reported). Statistical significance is not the same as biological significance.

Response: We thank the Reviewer for raising the important point that statistical significance does not equate to biological significance. We have noted this in the discussion of the text with specific reference to the significant association between preferences for facial masculinity and moral disgust (Lines 606-614).

5. I do have several other more minor questions / comments. These include asking how the 15 men used as stimuli were selected from the 37 men originally photographed; the origin of the '40 female and 40 male European faces'; correction of several typos; the (missing) distinction between 'mating success' and male-male competition in lines 109-110; missing column headings in Table 2; adding a key to the column headings in the supporting data Excel file. However, I won't elaborate on those or others now / yet, as the issues above are more pressing.

Response: We thank the Reviewer for raising these additional questions/comments, which are listed here with a response immediately below:

- how the 15 men used as stimuli were selected from the 37 men originally photographed:

Response: The men were selected at random from the full library of 37 men (photographed clean-shaven and bearded) using excel. A batch of five males were selected and used to create one composite clean-shaven and bearded. We created three composites in total. We have rewritten the methods (Lines 270-283) and we hope it is now easier to follow.

- the origin of the '40 female and 40 male European faces':

Response: The '40 female and 40 male European faces' the 40 female and 40 male faces that made up the composite from the masculinity transform were acquired from the online database 3d.sk. We have added this information to the methods (Lines 270-283).

- correction of several typos;

Response: We have been through the revised manuscript and have corrected the typos.

- the (missing) distinction between ‘mating success’ and male-male competition in lines 109-110:

Response: We have rewritten this section to explain that mating success was higher among men with facial hair than among men without facial hair during periods there were less unmarried women in the pool of potential mates and intra-sexual competition may be higher (Lines 107-113).

- missing column headings in Table 2:

Response: We have added the heading to Table 2.

- adding a key to the column headings in the supporting data Excel file.

Response: We have updated the Excel file and uploaded it into Dryad.

6. I still think there's a lot of value in the dataset and manuscript, and I regret not being more positive here. Usually, I feel much more positive about a manuscript upon second reading following revisions – the topic feels more familiar and digestion means that it's easier to follow the rationale and study. Consequently, I am concerned here where I still have reservations at this point. I hope I'm not being unfair.

Response: We understand and appreciate the Reviewers comments and we hope we have satisfied their concerns in our revised manuscript.